# Compounds activating VCP D1 ATPase enhance both autophagic and proteasomal neurotoxic protein clearance

Lidia Wrobel [1,2,8], Sandra M. Hill [1,2,3,8], Alvin Djajadikerta [1,2], Marian Fernandez-Estevez[1,2], Cansu Karabiyik[1,2], Avraham Ashkenazi [1], Victoria J. Barratt[1,2], Eleanna Stamatakou[1,2], Anders Gunnarsson[4], Timothy Rasmusson[5], Eric W. Miele[5], Nigel Beaton[6], Roland Bruderer[6], Yuehan Feng[6], Lukas Reiter[6], M. Paola Castaldi[5], Rebecca Jarvis[7], Keith Tan[7], Roland W. Bürli[7] & David C. Rubinsztein [1,2✉]

Enhancing the removal of aggregate-prone toxic proteins is a rational therapeutic strategy for a number of neurodegenerative diseases, especially Huntington's disease and various spinocerebellar ataxias. Ideally, such approaches should preferentially clear the mutant/misfolded species, while having minimal impact on the stability of wild-type/normally-folded proteins. Furthermore, activation of both ubiquitin-proteasome and autophagy-lysosome routes may be advantageous, as this would allow effective clearance of both monomeric and oligomeric species, the latter which are inaccessible to the proteasome. Here we find that compounds that activate the D1 ATPase activity of VCP/p97 fulfill these requirements. Such effects are seen with small molecule VCP activators like SMER28, which activate autophagosome biogenesis by enhancing interactions of PI3K complex components to increase PI(3)P production, and also accelerate VCP-dependent proteasomal clearance of such substrates. Thus, this mode of VCP activation may be a very attractive target for many neurodegenerative diseases.

[1] Department of Medical Genetics, Cambridge Institute for Medical Research, The Keith Peters Building, Cambridge Biomedical Campus, Hills Road, Cambridge CB2 0XY, United Kingdom. [2] UK Dementia Research Institute, University of Cambridge, Cambridge Institute for Medical Research, The Keith Peters Building, Cambridge Biomedical Campus, Hills Road, Cambridge CB2 0XY, United Kingdom. [3] Department of Psychiatry and Neurochemistry, Institute of Neuroscience and Physiology, The Sahlgrenska Academy at the University of Gothenburg, 413 45 Gothenburg, Sweden. [4] Biophysics SE, Discovery Sciences, R&D, AstraZeneca, Gothenburg, Sweden. [5] Discovery Biology, Discovery Sciences, R&D, AstraZeneca, Boston, MA, USA. [6] Biognosys AG, Wagistrasse 21, 8952 Schlieren, Switzerland. [7] Discovery UK, Neuroscience, BioPharmaceuticals R&D, AstraZeneca, Cambridge, UK. [8] These authors contributed equally: Lidia Wrobel and Sandra M. Hill. ✉email: dcr1000@cam.ac.uk

Many neurodegenerative diseases, including Alzheimer's, Parkinson's, and Huntington's diseases, manifest with the intracytoplasmic accumulation of toxic aggregate-prone proteins. Thus, understanding the pathways that impact their degradation may be of therapeutic utility. The pathways that have received the most attention in this context are the ubiquitin-proteasome-system (UPS) and lysosome-dependent macro-autophagy (henceforth autophagy)[1–3]. The UPS is the principal pathway for the clearance of short-lived and soluble misfolded proteins and polypeptides. However, while UPS-dependent clearance can be rapid, the narrow entrance of the proteasome requires that substrates are monomeric and unfolded, and thus precludes removal of oligomeric and higher-order species frequently found in the aforementioned diseases. Instead, such long-lived proteins, aggregate-prone proteins, and oligomeric protein complexes can be cleared by the autophagy-lysosome system. Hence, a strategy of activating both proteasomal and autophagy-dependent clearance of such proteins may be advantageous, as it would enable clearance of both monomeric and oligomeric/misfolded species, respectively.

In many of these diseases, it is desirable to be able to preferentially enhance clearance of the disease-causing protein species, including mutant and abnormally folded aggregate-prone species, while preserving the wild-type proteins. For example, in autosomal dominant conditions caused by polyglutamine expansions, like Huntington's disease and spinocerebellar ataxia type 3, the wild-type proteins have physiological and protective functions[4,5], and reducing the levels of the wild-type proteins in patients as a consequence of therapeutic attempts to lower the mutant protein may be deleterious[6].

The UPS is composed of three major components—the proteasome holoenzyme and various ubiquitin ligases and de-ubiquitinating enzymes. Degradation of target proteins is initiated by the recognition and sequential build-up of ubiquitin in various chains on substrates that enable their targeting to the proteasome for degradation. Proteasomal targeting of proteins that are a part of larger complexes, embedded in membrane structures, or tightly folded is often aided by the unfolding ability of the Valosin-containing protein (VCP, also called p97)[7,8]. VCP is also important for enabling proteasomal degradation of proteins containing ubiquitin fusion degradation (UFD) signals, and fusion proteins with a "non-removable" N-terminal ubiquitin (Ub) moiety[9].

In autophagy, substrates destined for degradation are enclosed in double-membraned autophagosomes, which ultimately fuse to lysosomes to enable the degradation of their contents by lysosomal proteases. One of the early key events in autophagy initiation is the production of PI(3)P by the Beclin 1 containing PI3K complex 1[10–12]. The PI(3)K complex 1 includes the core phosphoinositide 3-kinase regulatory subunit 4 (also known as VPS15), phosphatidylinositol 3-kinase catalytic subunit type 3 (also known as PI3KC3 or VPS34), Bcl2-interacting protein 1 (Beclin 1) and Beclin 1-associated autophagy-related key regulator (also known as ATG14). VPS34 is a class III phosphatidylinositol 3-kinase, which directly phosphorylates the 3-OH position of phosphatidylinositol (PI) thereby generating PI(3)P. PI(3)P is a lipid signaling molecule that marks the site for autophagosome formation and is essential for recruiting downstream factors such as WIPI2, ATG16, and LC3 for membrane expansion and substrate recruitment to the forming autophagosome[13,14]. We recently found that VCP inhibition impaired autophagosome formation by increasing the degradation of Beclin 1 and by independently compromising the assembly of the class III phosphatidylinositol 3-kinase complex[15].

Several autophagy-inducing molecules have been developed and tested in neurodegenerative disease models, where they can reduce the accumulation of toxic proteins and ameliorate signs of neurodegeneration[16,17]. Many of these compounds seem to modulate autophagy by inhibiting mTOR complex 1[16,17], a major signaling kinase regulating a wide array of cellular functions. Due to their broad action, molecules inhibiting mTOR may result in undesirable effects unrelated to autophagy, like immunosuppression[18]; thus, there is a need for developing alternative autophagy modulators acting independently of mTOR. We previously identified a small molecule SMER28 as one such mTOR-independent autophagy inducer, and while the exact mechanism has not been identified, studies have shown that SMER28 accelerates the autophagic degradation of harmful protein species that cause neurodegeneration both in vitro and in vivo[19–22]. Importantly, SMER28 induces autophagy in the brain of rats after intraperitoneal injection[22,23].

Upregulation of proteasome activity has also been considered a strategy for neurodegenerative diseases[24,25]. We wondered whether it was possible to enhance protein clearance via both pathways by targeting a single regulator, as this may enable more potent therapeutic effects. Here we have identified VCP/p97 as a target for SMER28. We show that SMER28 binds in the cleft formed between VCP's substrate N-terminal binding domain and ATPase domain 1 and that it stimulates its D1 ATPase activity. We further show that SMER28 increases PI(3)P production and induces autophagy in a VCP-dependent manner, thus providing data for a possible mechanism of action, in which SMER28 induces autophagy by targeting VCP and augmenting its role in autophagy initiation. Furthermore, we show that SMER28 enhances proteasome-mediated degradation of misfolded and aggregate-prone proteins in a VCP-dependent manner. Importantly, SMER28 lowers levels of mutant proteins while preserving the levels of the wild-type counterparts, thereby revealing desirable properties of this target to stimulate clearance of aberrant proteins by autophagy and UPS simultaneously.

## Results

**SMER28 induces autophagy flux and clears mutant neurodegeneration-causing proteins.** We previously reported that SMER28 induces autophagy[19]. To further validate the effects of SMER28 on the autophagic flux, we used a ratiometric assay that we have recently developed that generates a positive signal when there is lysosomal degradation of an autophagic substrate. The assay exploits the SRAI reporter, a tandem construct consisting of TOLLES (a blue fluorescent protein resistant to acid-denaturation and proteolysis) and YPet (a yellow fluorescent protein, which undergoes acid-denaturation and proteolysis in lysosomes)[26]. Delivery of the SRAI reporter to lysosomes causes degradation of YPet, which leads to a detectable shift in fluorescence of the tandem construct as the FRET-associated quenching of the TOLLES signal is relieved after YPet degradation (Supplementary Fig. 1a). To adapt this tool for autophagy, we fused the SRAI reporter to the N-terminus of LC3B, a classical autophagy marker protein that is present on both inner and outer membranes of autophagosomes and is delivered to lysosomes, where the LC3 on the inner membrane is exposed to proteases and the LC3 on the outer membranes is deconjugated from the membranes by ATG4 family members. By fusing the SRAI reporter to LC3B, we markedly increase the sensitivity of the reporter as a macroautophagy assay, as the ratio of blue to yellow fluorescence reflects the proportion of LC3B that is undergoing lysosomal degradation. Treatment of SRAI-LC3B-expressing HeLa cells with autophagy modulators, like the stimulators rapamycin and Torin 1 or the lysosomal inhibitor Bafilomycin A1, produces substantial shifts in the TOLLES:YPet ratio (Supplementary Fig. 1b, c), thus validating the assay. Treatment of SRAI-LC3B cells with SMER28 causes a clear dose-response

relationship (Fig. 1a), validating the autophagy-inducing effects of the compound.

Previous studies have shown that treatment with SMER28 decreases the levels of disease-causing proteins that are autophagy substrates, including mutant huntingtin exon 1 (aggregate levels in wild-type but not autophagy-null cells) and the A53T α-synuclein mutant that causes Parkinson's disease, both in cultured cells and animal models[19–22]. We confirmed that SMER28 could reduce the levels of mutant huntingtin expressed at endogenous levels in knock-in mouse striatal cell lines homozygous for either mutant (Q111/Q111) or wild-type huntingtin (Q7/Q7)[27], as well as in patient-derived fibroblasts heterozygous for the mutation. After 24 h of treatment with SMER28 we observed a significant decrease in the levels of full-length mutant polyQ111 HTT, but not wild-type HTT in the striatal cells (Fig. 1b, c). Likewise, SMER28 significantly reduced the levels of polyQ-expanded HTT in different HD fibroblast lines using an antibody that detects the mutant but not the wild-type protein (Fig. 1d; Supplementary Fig. 1d). Importantly, SMER28 treatment did not significantly alter the levels of wild-type huntingtin in control fibroblasts detected with specific anti-huntingtin antibody (Fig. 1e). Similarly, SMER28 selectively decreased the levels of polyQ-expanded Ataxin 3 but not the wild-type Ataxin 3 in SCA3 patient-derived fibroblasts (Fig. 1f). Next, we analyzed the protein aggregation burden in unaffected control and HD patient fibroblasts with mutant polyQ80 HTT upon treatment with SMER28. Protein aggregation was assessed using the Proteostat dye, which recognizes aggregates from a broad range of protein substrates that accumulate when the proteasome is inhibited upon MG132 treatment[28] (Fig. 1g). Although control cells displayed an even staining with the dye in basal conditions, cells derived from HD patient showed obvious aggregate staining, which was significantly reduced when cells were treated with SMER28 (Fig. 1g, h).

**SMER28 binds VCP and increases VCP D1 ATPase activity**. To identify molecular targets of SMER28, we used a reverse competition pulldown assay, whereby protein lysates were pre-treated with either DMSO or SMER28, followed by an enrichment step with beads carrying a linker-modified SMER28 analog (Supplementary Fig. 1e). The proteins enriched by SMER28 on beads were subsequently assessed by mass spectrometry (Supplementary Data 1). Of the two most competed targets across two replicates, aldehyde dehydrogenase (ALDH1) binding is consistent with the fact that the quinazoline core unit of SMER28 is a prominent substrate for aldehyde oxidase (see e.g., ref. [29]). In contrast, VCP (also known as p97) was considered a unique hit (Fig. 2a), and VCP showed the highest fold-change (Supplementary Data 1). We considered VCP a plausible target as inhibition of its activity has been shown to compromise both autophagosome biogenesis and maturation[15,30,31].

Binding of SMER28 to VCP was confirmed using surface plasmon resonance (SPR) and a concentration-dependent binding of SMER28 with sub-micromolar affinity ($K_d$ ~750 ± 250 nM) was demonstrated (Supplementary Fig. 1f). The affinities determined for the two reference ligands ADP and ML240 ($K_d$ ~800 ± 100 nM and $K_d$ = 230 ± 30 nM, respectively) are in good agreement with previously reported values[32]. Notably, the lower $R_{max}$ determined for SMER28 relative to the two control compounds is a direct consequence of its lower molecular weight.

In order to verify this interaction in cells in an orthogonal fashion, we confirmed the interaction between SMER28 and VCP in cells using a Drug Affinity Responsive Target Stability assay (DARTS)[33], where binding of SMER28 protects VCP from proteolytic digestion by pronase (Fig. 2b). Thus, we showed that SMER28 binds VCP in vitro and in cultured cells.

VCP exists as a hexamer, and each monomer consists of two ATPase domains, D1 and D2, joined by a linker region[7,34]. The binding of SMER28 to VCP caused a slight but significant and dose-dependent increase in VCP ATPase activity (Fig. 2c, Supplementary Fig. 1g, h). The effect of SMER28 on VCP ATPase activity was blocked during co-treatment with VCP inhibitors DBeQ, NMS873, and CB-5083[35,36] (Fig. 2c, Supplementary Fig. 1h). To determine whether SMER28 binding to VCP selectively affects the individual ATPase domains, we purified Walker B mutants of VCP that have ATPase activity restricted to the D1 (E578Q) or D2 domains (E305Q)[37]. We found that SMER28 could stimulate in vitro ATPase activity of the D1 domain, but not D2 domain (Fig. 2d, e). We used VCP inhibitors as control, to show that CB-5083 targets only the D2 domain and NMS873 targets both[32] (Fig. 2d, e). VCP mutations associated with multisystem proteinopathy (IBMPFD) exhibit increased D2 ATPase activity[32,38,39]. Interestingly, SMER28 did not cause a further increase in the activity of one such mutation, VCP-R155H (Supplementary Fig. 1i). The D1 domain of VCP was previously reported to be important for the VCP hexamer formation[40]. To determine if SMER28 could alter VCP hexamerisation, we treated HeLa cells with SMER28 and VCP inhibitor as control and analyzed samples using native gel conditions (Fig. 2f, g). VCP inhibitor decreased the levels of VCP hexamer as expected, but SMER28 did not cause any significant change (Fig. 2f, g). Thus, SMER28 binds to VCP and increases its D1 ATPase activity without affecting hexamer formation.

To further support the relationship between the SMER28 interaction with VCP and its effect on autophagy, we tested a panel of structural analogs (analogs A–G; Supplementary Fig. 2a). We found that analogs A and C-G could bind VCP, as reflected by the ability to protect VCP from degradation in the DARTS assay, whereas analog B could not (Supplementary Fig. 2b–d). Correspondingly, analogs A and D-G could stimulate VCP ATP hydrolysis in a manner similar to SMER28, whereas the non-binding compound B and compound C did not display an effect (Supplementary Fig. 2e), supporting the link between VCP binding and modulation of its activity. Additionally, the ability of SMER28 and its analogs to protect VCP from degradation in the DARTS assay (binding) correlated with the capacity to induce the formation of LC3-positive autophagosomes (Supplementary Fig. 2f, g) and to decrease the levels of the autophagic substrate mutant α-synuclein (A53T, Supplementary Fig. 2h). Analog C was an exception, as it showed the ability to bind VCP without affecting either ATPase activity or α-synuclein levels, and only had a modest effect on autophagosome formation. This indicates that both binding and modulation of VCP ATPase activity are necessary for proper autophagy induction. Thus, SMER28 interacts with VCP in vitro and in cells, and modulation of VCP ATPase activity in the D1 domain upon binding correlates with the autophagy-inducing properties of SMER28.

To further understand the impact of SMER28 binding to VCP, we employed the limited proteolysis-coupled mass spectrometry (LiP-MS) approach, which exploits peptide level resolution data to predict compound binding sites. The previously established approach of using the top three ranking peptides to triangulate a binding site based upon the peptide's center of mass[41] was used. Typically, peptides would be ranked based upon LiP score, which is weighted based on several parameters to differentiate the target likelihood between proteins. As we had identified the target of SMER28, the approach was modified here to rank peptides from VCP based upon their dose-response correlation (Supplementary Fig. 2i), which is the largest factor in calculating LiP scores. Of the 467 peptides (including modifications) identified in the experiment from VCP, only three showed a very high correlation ($R^2 > 0.9$) to SMER28 treatment (Supplementary Fig. 2j). These peptides also showed the lowest predicted EC50 values. Such

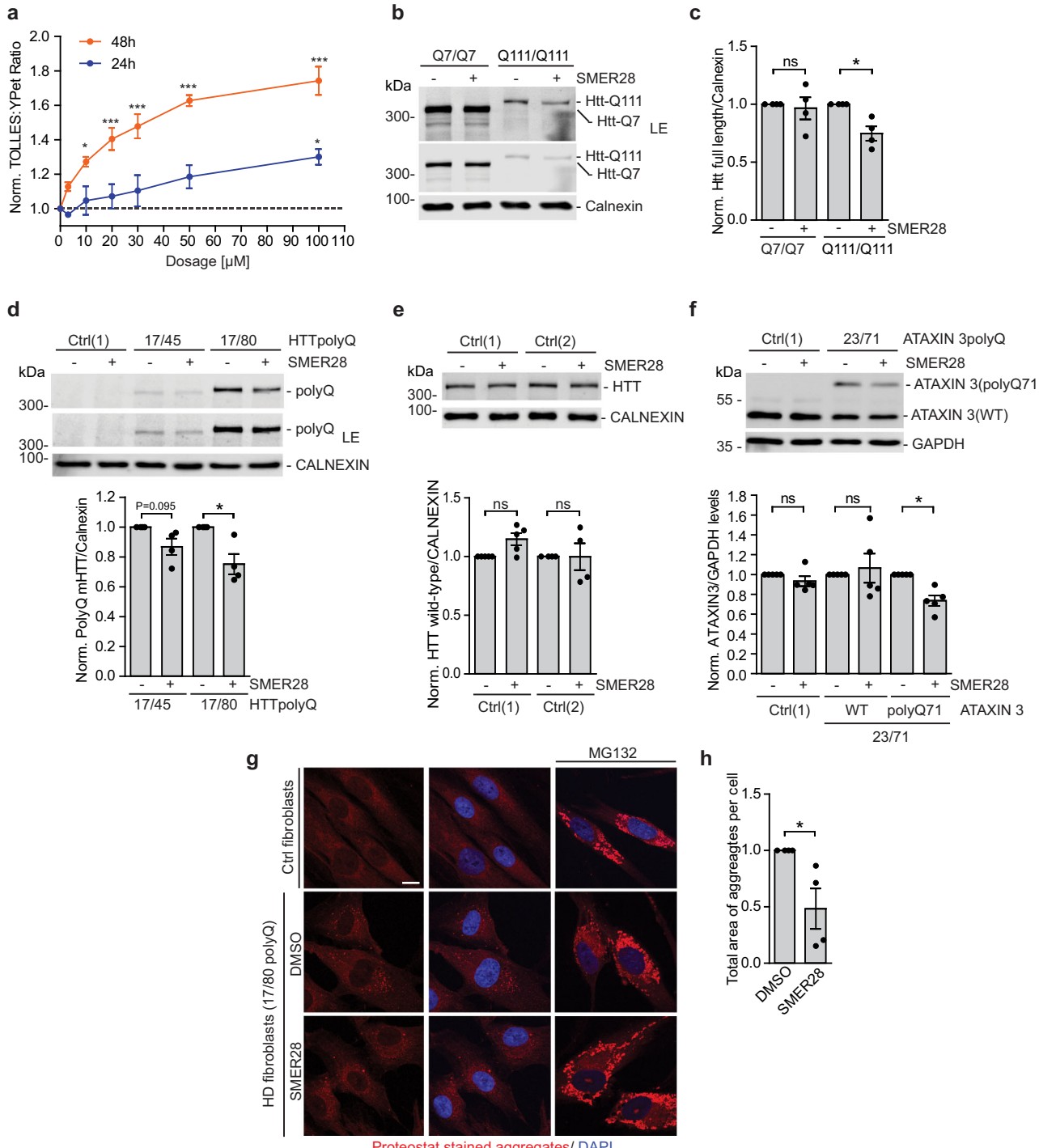

**Fig. 1 SMER28 induces autophagy flux and clears mutant neurodegeneration-causing proteins. a** HeLa cells expressing SRAI-LC3B were treated with a series of SMER28 concentrations (3 μM, 10 μM, 20 μM, 30 μM, 50 μM, and 100 μM) or DMSO for 24 h or 48 h and analyzed by FACS; one-way ANOVA for 48 h: *P* < 0.0001 (10 μM *P* = 0.0097, 20 μM *P* = 0.003, 30–100 μM *P* < 0.0001) and for 24 h: *P* = 0.0264 (100 μM *P* = 0.0187) with post hoc Tukey test, *n* = 3. **b**, **c** Mouse striatal cells expressing wild-type (Q7/Q7) or mutant (Q111/Q111) huntingtin were treated with 20 μM SMER28 for 24 h. Cells were lysed and analyzed by western blotting with anti-huntingtin antibody, *n* = 4; quantification in **c** (Q7/Q7 *P* = 0.7384, Q111/Q111 *P* = 0.0262). **d**, **e** Control (Ctrl) and Huntington's Disease (HD) fibroblasts expressing mHTT-polyQ45 or mHTT-polyQ80 were treated with 20 μM SMER28 for 24 h. Cells were lysed and analyzed by western blotting with anti-polyQ (**d**, *n* = 4; HD17/45 *P* = 0.0953; HD17/80 *P* = 0.0355) or anti-huntingtin antibody (**e** *n* = 5).
**f** Control (Ctrl) and Spinocerebellar ataxia type 3 (SCA3) fibroblasts were treated with 30 μM SMER28 for 24 h. Cells were lysed and analyzed by Western blotting with anti-Ataxin 3 antibody, *n* = 5; Ctrl1 *P* = 0.2491, WT *P* = 0.6842, polyQ *P* = 0.0075. **g**, **h** Control or HD fibroblast cells expressing mHTT-polyQ80 were treated with DMSO, 20 μM SMER28 or 1 μM MG132 as control. Cells were fixed and stained with Proteostat dye. Total area of aggregates puncta was quantified in **h**; *n* = 4, *P* = 0.497; scale bar = 10 μm. Bar graphs data presented as normalized mean ± SEM. *\*P* < 0.05, *\*\*P* < 0.001, one sample *t* test unless stated otherwise; LE long exposure, WT wild-type, HTT, Huntingtin, Ctrl control cells, ns not significant. Source data are provided as a Source Data file.

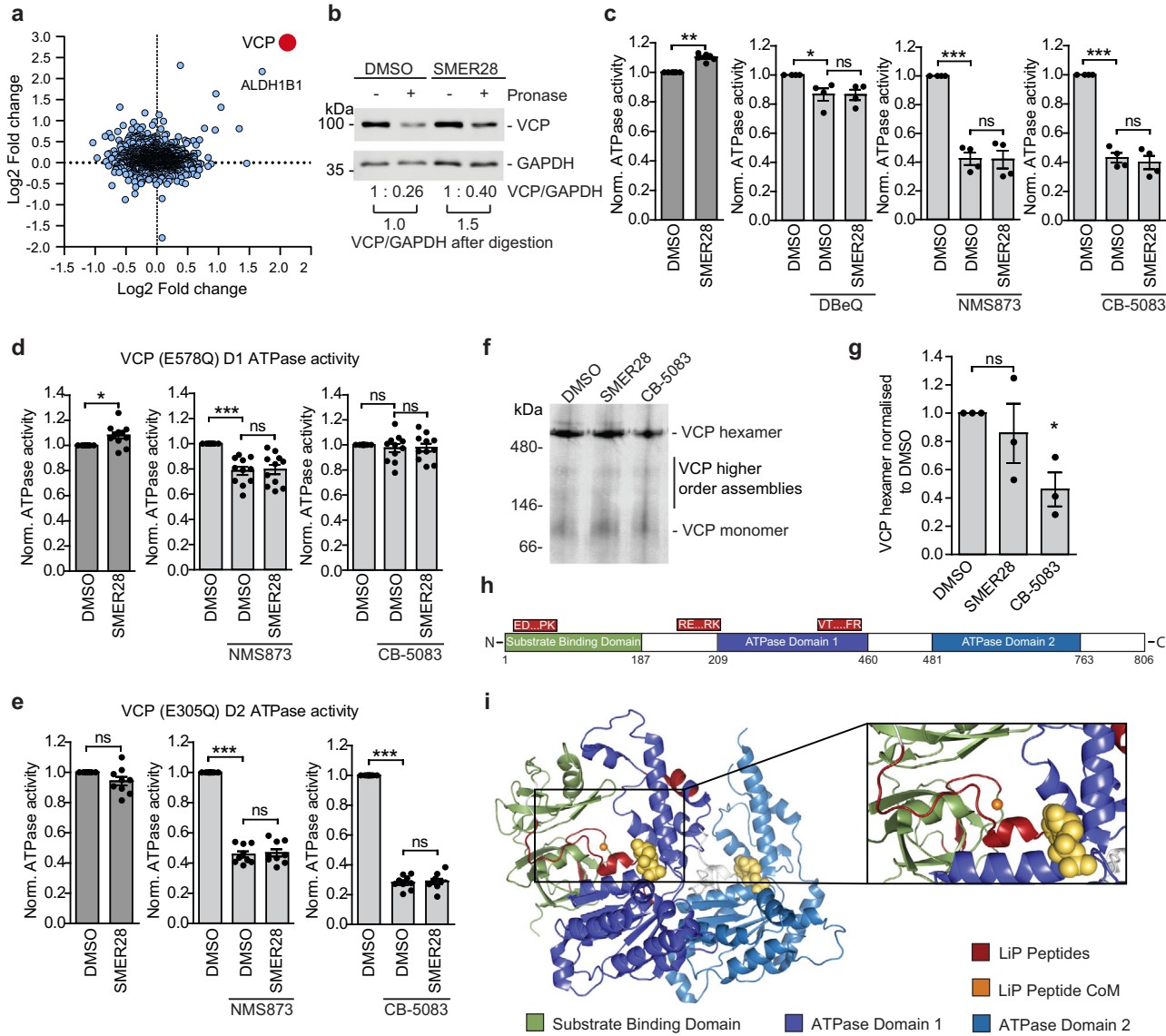

**Fig. 2 SMER28 binds VCP and increases VCP D1 ATPase activity. a** SMER28 interactors tested in a reverse competition experiment with free compound at 800 µM. Interactors with a log2-fold-change >2 are labeled in red and were considered significant. Data point size correlates with a number of unique peptides. **b** VCP levels after pronase digestion in DARTS assay in lysates from HeLa cells treated with 20 µM SMER28 for 1 h before lysis. VCP/GAPDH ratio normalized to non-digested samples and ratio in treated samples compared to DMSO control. **c–e** in vitro ATPase activity of wild-type VCP (**c** $n = 7$), VCP-E578Q mutant (**d** $n = 11$), and VCP-E305Q mutant (**e** $n = 9$) upon addition of 20 µM SMER28 with and without the addition of VCP inhibitors: 10 µM DBeQ, 10 µM NMS873 or 2 µM CB-5083 after 60 min incubation at 37 °C; statistical analysis of DMSO vs SMER28: one sample $t$ test (**c** $P = 0.0002$; **d** $P = 0.0115$; **e** $P = 0.0772$); statistical analysis of DMSO vs SMER28 in presence of VCP inhibitors: one-way ANOVA with post hoc Tukey test (for details see Source Data file). **f**, **g** HeLa cells were treated with 20 µM SMER28 or 5 µM CB-5083 for 6 h, followed by Blue-Native-PAGE electrophoresis; $n = 3$, quantification in **g**; one-way ANOVA: $P = 0.0489$ with post hoc Tukey test. **h** Linear representation of VCP protein with SMER28 binding peptides from Supplementary Fig. 2i represented on top in red. **i** Peptides from Supplementary Fig. 2i mapped (red) onto one subunit of VCP with the center of mass (CoM, orange) of the three peptides indicated. Known domains of VCP (green, purple, and blue) and ADP (yellow) are indicated. Bar graphs data presented as normalized mean ± SEM. *$P < 0.05$, **$P < 0.001$, ***$P < 0.0001$; ns not significant. Source data are provided as a Source Data file.

peptides have been demonstrated in previous experiments to lie closest to small molecule binding sites. Using the aforementioned triangulation approach, these three top LiP peptides were mapped (red) onto the structure of a single subunit of VCP (Fig. 2h, i, pdb: 5ftk) and the center of mass was calculated (orange dot) in 3D space. This technique predicts that SMER28 binds in the cleft formed between VCP's substrate binding domain (green) and ATPase domain 1 (purple). Collectively, these data provide strong evidence that SMER28 binds in the VCP cleft between the substrate binding domain and ATPase domain 1 and increases its D1 ATPase activity.

**Autophagy induction by SMER28 depends on VCP and PI3K activity.** Consistent with the role of VCP in early autophagy-inducing events[15], SMER28-induced a significant increase in PI(3)P production (Fig. 3a, c) which was comparable with the increase in PI(3)P observed upon nutrient depletion (Fig. 3b). This was dependent on VPS34 kinase activity as well as the ATPase activity of VCP, as the increase in PI(3)P production by SMER28 (or starvation) was blocked upon co-treatment with the VPS34 inhibitor IN1[42] (Fig. 3b, c) or with VCP inhibitors CB-5083, NMS873 or DBeQ (Fig. 3c, d). Additionally, siRNA-mediated VCP knockdown impaired SMER28-induced production of PI(3)P, whereas

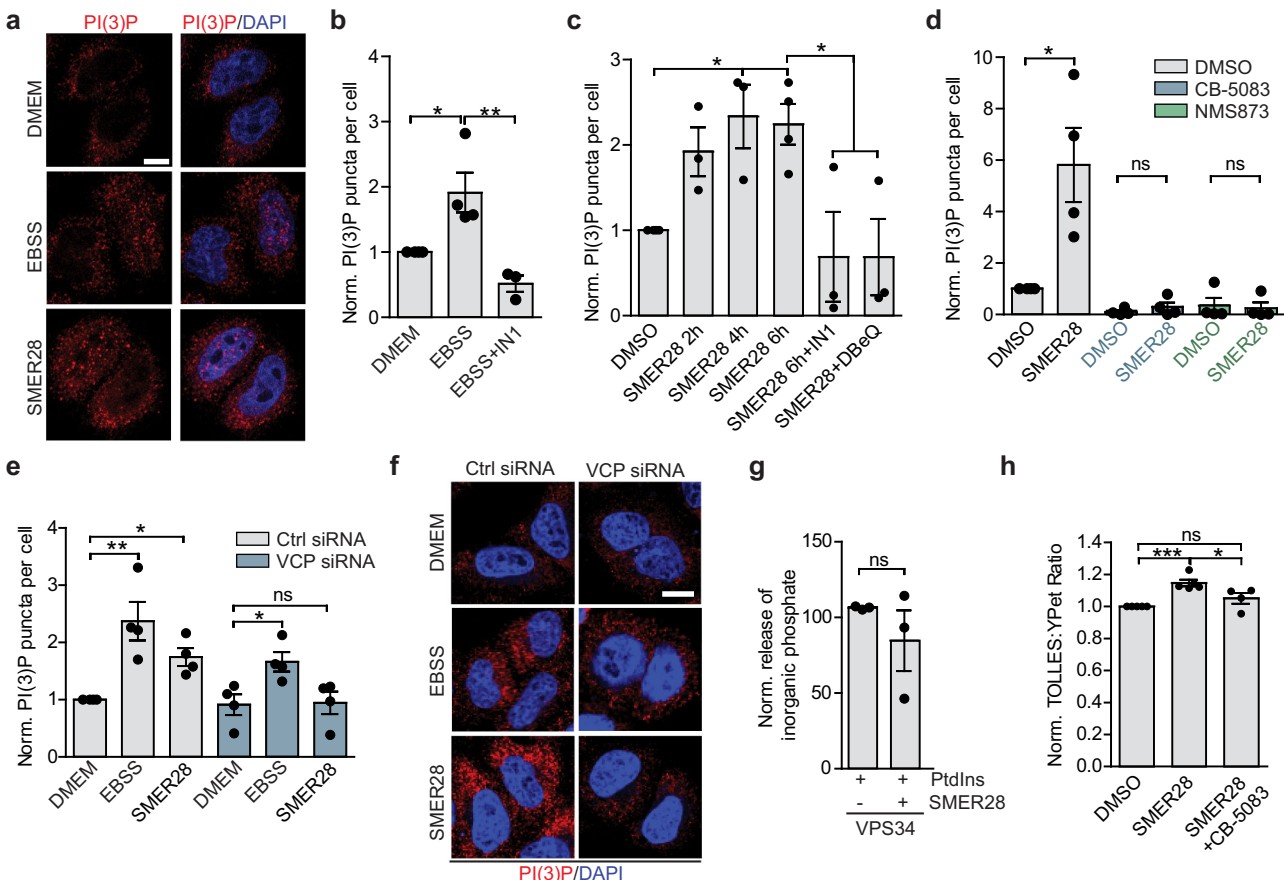

**Fig. 3 Autophagy induction by SMER28 depends on VCP and PI3K activity. a–c** PI(3)P puncta in HeLa cells in basal (DMEM), starvation (EBSS 1 h; **a**, **b**) and in basal conditions treated with 20 μM SMER28 (**a**, **c**) or with or without 1 μM VPS34 inhibitor (IN1) or 10 μM DBeQ for indicated time points (**b**, **c**); PI(3)P puncta per cell normalized to control; $n = 4$ representative images in **a**; quantification in **b** (one-way ANOVA: $P = 0.004$ with post hoc Tukey test; EBSS $P = 0.0255$, EBSS + IN1 $P = 0.0039$) and in **c** (for SMER28 treatment alone: one-way ANOVA: $P = 0.0079$ with post hoc Tukey test, 4 h SMER28 $P = 0.0131$, 6 h SMER28 $P = 0.0126$; for SMER28 6 h treatment with or without inhibitors: one-way ANOVA: $P = 0.0153$ with post hoc Tukey test, SMER28 + IN1 $P = 0.0284$, SMER28 + DBeQ $P = 0.028$). **d** PI(3)P puncta in cells treated with 20 μM SMER28 for 6 h with or without 5 μM CB-5083, 10 μM NMS873; $n = 4$; two-tailed paired Student's $t$ test; DMSO vs. SMER28 $P = 0.0444$. **e**, **f** PI(3)P puncta in control and VCP knockdown HeLa cells in basal (DMEM), starvation (EBSS 2 h) and in basal conditions treated with 20 μM SMER28 for 4 h; $n = 4$; representative images in **f**; quantification in **e** (for control siRNA cells: one-way ANOVA $P = 0.0022$ with post hoc Tukey test; for VCP knockdown cells: one-way ANOVA $P = 0.0294$ with post hoc Tukey test). **g** Purified VPS34 was incubated in the presence of phosphatidylinositols (PtdIns) and ATP with 20 μM SMER28 for 20 min. The levels of inorganic phosphate were measured; $n = 3$; paired two-tailed Student's $t$ test. **h** SRAI-LC3B cells were treated with 20 μM SMER28 in the presence or absence of 0.3 μM CB-5083 for 24 h, followed by FACS analysis, $n = 5$; one-way ANOVA: $P = 0.01$ with post hoc Tukey test (DMSO vs. SMER28 $P = 0.0008$, CB-5083 $P = 0.0213$). Data presented as normalized mean ± SEM, $*P < 0.05$, $**P < 0.001$, $***P < 0.0001$; scale bar = 10 μm; ns not significant. Source data are provided as a Source Data file.

starvation-induced increase in PI(3)P was still observed in these cells, although much reduced, in line with previous observations (Fig. 3e, f, Supplementary Fig. 3a)[15]. SMER28 analogs that bound VCP (Supplementary Fig. 2c) increased PI(3)P levels, similar to SMER28 (Supplementary Fig. 3b). Consistent with our autophagy-related data above, the non-binding analog B did not cause any discernible PI(3)P induction, thus strengthening the correlation of SMER28 binding to VCP with its autophagy-inducing potential (Supplementary Fig. 3b). The increase of PI(3)P was not due to SMER28 affecting VPS34 directly, as SMER28 did not show significant activity against VPS34 in a kinase screen (Supplementary Data 2), nor did it affect VPS34 intrinsic ATPase activity (Supplementary Fig. 3c) or its kinase activity in vitro (Fig. 3g, Supplementary Fig. 3d). The induction of autophagy by SMER28, measured as an increase in LC3-II formation in the presence of Bafilomycin A1, was also dependent on VCP, as this effect was significantly reduced when cells were co-treated with the VCP inhibitor DBeQ (Supplementary Fig. 3e) and CB-5083 or NMS873

(Supplementary Fig. 3f). Furthermore, SMER28-mediated increase in autophagic flux, as measured by the SRAI-LC3B reporter, was blocked by co-treatment with VCP inhibitor (Fig. 3h). VCP is also involved in the endoplasmic reticulum associated degradation (ERAD), therefore we asked if SMER28 treatment might induce selective degradation of the ER (ER-phagy)[43,44]. SMER28-induced clearance of the general autophagy receptor p62, but it did not change the levels of ER-phagy specific receptor TEX264 after up to 9 h of treatment (Supplementary Fig. 3g). While we cannot exclude that a longer period of SMER28 treatment may enhance ER-phagy, 9 h starvation-induced clearance of both receptors (Supplementary Fig. 3g). Thus, SMER28 induces autophagy by stimulation PI(3)P formation in a VCP-dependent manner.

Testing SMER28 in an in vitro kinase panel revealed PI4K, IRAK4, and PIK3C as potential alternative targets (Supplementary Data 2), and raised the question of whether inhibition of any of these kinases could explain part of the autophagy-inducing effect by SMER28. However, subsequent tests with selected

kinases together with structural analogs of SMER28 (Supplementary Data 3) showed no clear correlation between potential kinase inhibition and properties of autophagy induction. An inhibition of PIK3C could induce autophagy, mediated by signaling through the AKT/mTOR axis[45]. We found no effect on AKT activation in cells upon treatment with SMER28 (Supplementary Fig. 3h), and these data together with previously published data show that SMER28 treatment has no effect on downstream mTOR signaling[19,46] further exclude the possibility of PIK3C as a target for the autophagy-inducing mechanism of SMER28.

**SMER28 and NW1030 both induce autophagy by increasing PI(3)P production.** A recent study identified the VCP binding compound NW1030 and showed that, similarly to SMER28, it binds to a cleft between the N-terminal domain and D1 ATPase domain of VCP to selectively stimulate D1 ATPase activity[47] and that this increased ATPase activity potentially correlates with an increase in autophagic flux. Interestingly, two of the three peptides identified by LiP-MS to interact with SMER28 overlap with two peptides identified as interactors of NW1030 (Fig. 4a, bottom, multicolored). Mapping the predicted LiP peptide CoM (orange) onto the VCP structure (pdb: 5dyi) from this previous study (including peptide identifications) demonstrates that the peptides flank the same cleft predicted to be the binding site of SMER28 (compare Fig. 2i and Supplementary Fig. 4a), which one would expect to be consistent between compounds that have similar phenotypic effects.

However, no mechanistic detail for NW1030 action was provided in this study. We could confirm the autophagy-inducing phenotype of this compound by measuring LC3 puncta in cells (Supplementary Fig. 4b). Ability of both NW1030 and SMER28 to increase autophagosomes (as measured by LC3 puncta) was not additive, suggesting that they act in a similar manner (Supplementary Fig. 4b). Consistent with our data for SMER28, treatment with NW1030 decreased the levels of full-length mutant polyQ111 HTT, but not wild-type HTT in the striatal cells (Supplementary Fig. 4c). Furthermore, the increase in LC3 puncta and LC3 lipidation levels upon treatment with NW1030 was dependent on VCP activity, as it was blocked when cells were co-treated with VCP inhibitors CB-5083 and NMS873 (Fig. 4b, Supplementary Fig. 4d, e). Moreover, consistent with our data for SMER28, NW1030 increased the levels of PI(3)P, which was dependent on VCP ATPase activity (Fig. 4c, Supplementary Fig. 4f). Recruitment of WIPI2 and ATG16 to the phagophore membrane enable LC3 lipidation and WIPI2 and ATG16L1 puncta are increased after nutrient depletion, which stimulates autophagy induction (Fig. 4d). Likewise, both SMER28 and NW1030 induced accumulation of WIPI2 and ATG16 in a way that was dependent on VCP ATPase activity as it was blunted by co-treatment with CB-5083 (Fig. 4e, f; Supplementary Fig. 4g). Thus, both SMER28 and NW1030 induce PI(3)P production and increase the recruitment of early autophagy markers in a VCP-dependent manner, further strengthening our hypothesis that the autophagy-inducing capabilities of SMER28 are linked to its effect on VCP.

**SMER28 enhances VCP stimulation of PI3K complex formation to induce autophagy in a Beclin 1-dependent manner.** While VCP has an established role in stabilizing Beclin 1 levels[15], treatment with SMER28 did not have any significant effect on these levels in basal growth conditions (Fig. 5a), as well as when lysosomal degradation was blocked by Bafilomycin A1 treatment (Fig. 5b). As a control condition, starvation (HBSS) showed a slight stabilization of Beclin 1 (Fig. 5a). Consistently, we did not observe an effect on VCP levels itself upon SMER28 treatment (Fig. 5a, b), further demonstrating that the effects of SMER28 are

not explained by stabilization of the target protein itself, and thus it is not regulating the autophagic degradation of VCP complexes[48]. However, autophagy induction by SMER28 was dependent on Beclin 1, as no effects were detected in Beclin 1 knockout cells (Fig. 5c, d). While the control cells display a significant increase in LC3-II levels after 8 h of treatment with SMER28, no such response could be observed in the Beclin 1 knockout cells (Fig. 5c, d). Similarly, autophagy induction by NW1030 was dependent on Beclin 1, as treatment with NW1030 increased LC3 puncta in control and not in Beclin 1 knockout cell (Fig. 5e, Supplementary Fig. 5a).

Recently, we reported that VCP regulates the assembly and activity of the PI(3)P-producing PI3K complex[15]. In agreement with this role for VCP, the addition of SMER28 increased the binding of VCP to ATG14L in cells, as well as the interaction between ATG14L and the other of the PI3K complex components VPS15, VPS34, and Beclin 1 (Fig. 5f, g). As a control, the addition of the VCP inhibitor DBeQ displayed the opposite effect. An increase in PI3K complex assembly upon SMER28 treatment was dependent on VCP ATPase activity as it was abolished by DBeQ (Supplementary Fig. 5b). The effect of SMER28 on PI3K complex assembly was confirmed in vitro (Fig. 5h, i; Supplementary Fig. 5c), as SMER28 together with VCP caused ATG14L to also interact more strongly with VPS15, VPS34 and Beclin 1. The addition of SMER28 alone to unassembled PI3K components had no effect, indicating that its actions are mediated via VCP (Fig. 5h, i). When we tested the effects of SMER28 on binary interactions of VCP with PI3K complex I components, we found that SMER28 had the strongest effect on the interaction between VCP and the kinase VPS34 (Supplementary Fig. 5d, e). Thus, our data suggest that SMER28 increases the VCP interaction with the entire PI3K complex I and also increases the formation of this complex. Taken together, these findings provide an explanation for how SMER28 increases PI(3)P formation and drives autophagy in a VCP-dependent manner.

**VCP activation enhances flux through UPS.** As VCP enables the clearance of certain UPS substrates[7,8], we considered the possibility that increasing VCP D1 ATPase activity by treatment with SMER28 could boost the removal of such proteins. We used cells stably expressing Ub-G76V-GFP, a ubiquitin fusion degradation (UFD) reporter that is degraded by the proteasome, which is dependent on VCP but not various other UPS upstream signals[9]. Under basal conditions, Ub-G76V-GFP is rapidly degraded, but accumulates if proteasome activity is impaired[49]. SMER28 treatment accelerated the degradation capacity of the UPS as it further decreased the levels of Ub-G76V-GFP reporter (Fig. 6a, Supplementary Fig. 5f). As expected, co-treatment with proteasome inhibitor MG132 stabilized the reporter levels (Fig. 6a) indicating that observed effect of SMER28 depends on enhanced UPS flux. Consistent with our previous data using the DARTS assay (Supplementary Fig. 2b–h) analogs A and D-G, but not B and C, enhanced the degradation of Ub-G76V-GFP reporter (Fig. 6b). In agreement with the key function of VCP in the UPS, treatment with VCP ATPase activity inhibitors, NMS873 or CB-5083, or siRNA-mediated VCP knockdown caused pronounced accumulation of Ub-G76V-GFP reporter (Fig. 6c; Supplementary Fig. 5g, h) and in these cells SMER28-mediated degradation was blocked (Fig. 6c, d), indicating that SMER28 increases UPS flux in the VCP-dependent manner. VCP binds various cofactors, which determine its cellular localization and substrate specificity[50]. One of the most crucial cofactors is the UFD1L/NPL4 (UN) heterodimer, an essential complex for many VCP-dependent processes[51,52]. Consistent with its role in the binding of the polyubiquitin chains, siRNA-mediated knockdown of either UFD1L or NPL4 impaired SMER28-induced

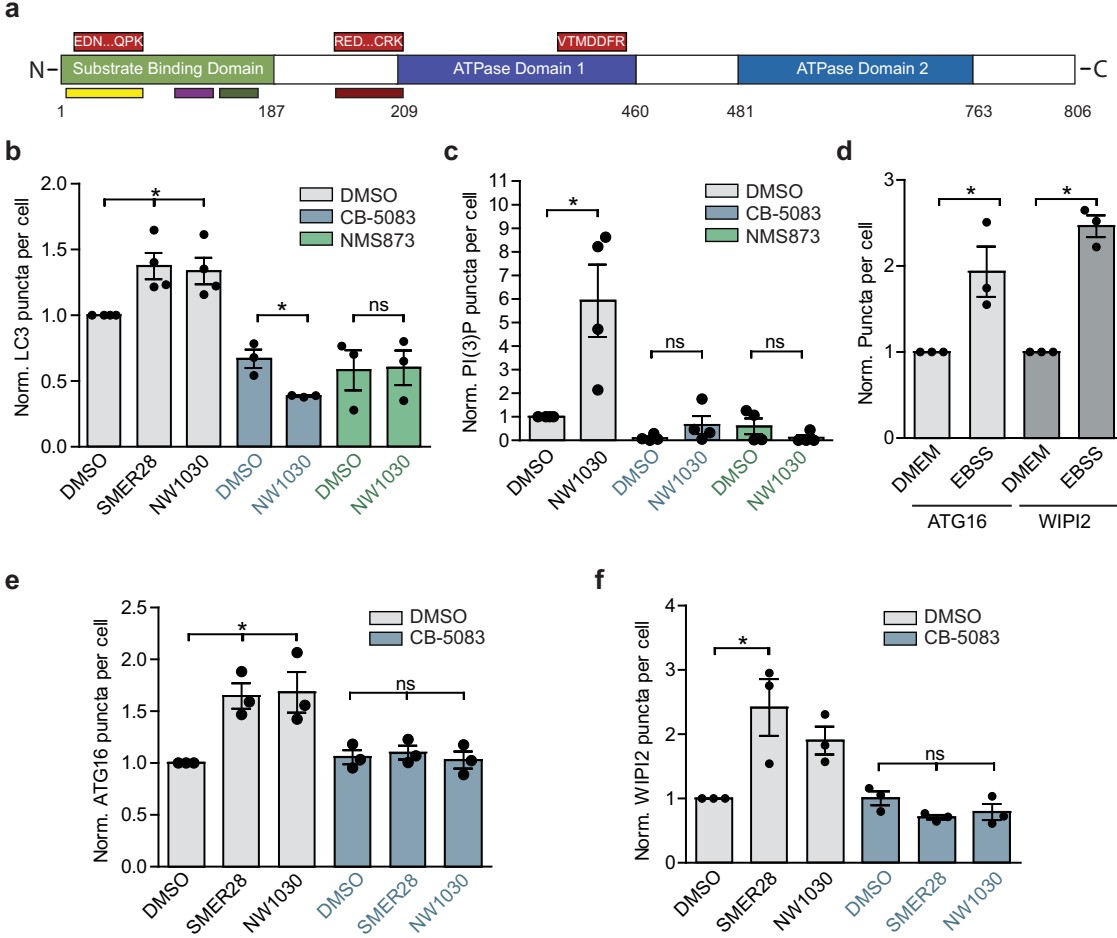

**Fig. 4 SMER28 and NW1030 both induce autophagy by increasing PI(3)P production. a** Linear representation of VCP protein with SMER28 binding peptides from Fig. 2 represented on top (red) and NW1030 binding peptides[47] indicated at the bottom (multicolor). **b** Number of LC3 puncta per cell normalized to DMSO control in cells treated with 20 μM SMER28, 10 μM NW1030 with or without 5 μM CB-5083 or 10 μM NMS873 for 8 h in presence of 400 nM BafA1, $n = 4$, representative images in Supplementary Fig. 4e; for SMER28 and NW1030 treatment alone: one-way ANOVA $P = 0.0197$ with post hoc Tukey test, NW1030 $P = 0.0321$, SMER28 $P = 0.0191$; for treatment in the presence of VCP inhibitors paired two-tailed Student's $t$ test. **c** PI(3)P puncta in cells treated with 10 μM NW1030 with or without 5 μM CB-5083 or 10 μM NMS873 for 4 h; $n = 4$; paired Student's $t$ test, NW1030 $P = 0.049$; representative images in Supplementary Fig. 4f. **d–f** ATG16 and WIPI2 puncta formation in HeLa cells in basal (DMEM) and starvation (EBSS 2 h) in **d** or in cells treated with 20 μM SMER28, 10 μM NW1030 with or without 5 μM CB-5083 for 4 h in **e**, **f**; $n = 3$; representative images in Supplementary Fig. 4g; quantification in **d** (paired one-tailed Student's $t$ test, ATG16 $P = 0.0429$, WIPI2 $P = 0.0037$); quantification in **e** (for SMER28 and NW1030 treatment alone: one-way ANOVA $P = 0.0189$ with post hoc Tukey test, SMER28 $P = 0.0248$, NW1030 $P = 0.0199$; for treatment in the presence of VCP inhibitors: one-way ANOVA $P = 0.788$ (ns)); quantification in **f** (for SMER28 and NW1030 treatment alone: one-way ANOVA $P = 0.0326$ with post hoc Tukey test, SMER28 $P = 0.0219$; for treatment in the presence of VCP inhibitors: one-way ANOVA $P = 0.171$ (ns)). Data presented as normalized mean ± SEM, $*P < 0.05$, $**P < 0.001$, $***P < 0.0001$; scale bar = 10 μm; ns not significant. Source data are provided as a Source Data file.

degradation of Ub-G76V-GFP reporter (Fig. 6e, Supplementary Fig. 5i). SMER28 binding to VCP could potentially alter the association of VCP with UFD1L/NPL4, influencing the recruitment of the ubiquitinated substrates. However, when we assessed the binding between VCP and UFD1l/NPL4 in the presence or absence of SMER28 using in vitro binding assay (Supplementary Fig. 5j) and VCP immunoprecipitation (Supplementary Fig. 5k), we found that SMER28 did not significantly alter VCP and UFD1l/NPL4 binding. Moreover, intrinsic proteasome activity was not affected by treatment with SMER28 (Fig. 6f, Supplementary Fig. 5l, m), further supporting that SMER28 enhances the targeting of proteasome substrates for degradation upstream of the proteasome.

**SMER28 enhances degradation of short-lived misfolded proteins through UPS in VCP and UFD1L/NPL4 dependent manner.** VCP has a well-established role in the degradation of aberrant/misfolded proteins by aiding their release from cellular

structures or large protein complexes[7,8]. To test the role of VCP in the degradation of aberrant proteins we examined the levels of puromycin-induced misfolded proteins in the cytosol upon SMER28 treatment. Puromycin is incorporated into nascent chains and the resulting misfolded proteins are released from ribosomes and their levels can be detected with puromycin-specific antibodies[53]. Treatment with SMER28 significantly decreased the accumulation of misfolded proteins in HeLa, SH-SY5Y, and mouse primary cortical neurons (Fig. 7a, b; Supplementary Fig. 6a), soluble on gels. These were not caused by the global change in protein translation, as the levels of properly folded nascent chains labeled with AHA (L-azidohomoalanine)[54] were not affected by SMER28 treatment (Fig. 7c; Supplementary Fig. 6b). Furthermore, the addition of SMER28 did not affect the degradation of endogenous SCD1, an ER-associated degradation substrate[55], or short-lived p53 (Supplementary Fig. 6c–e), and did not induce the unfolded protein response of the ER as measured

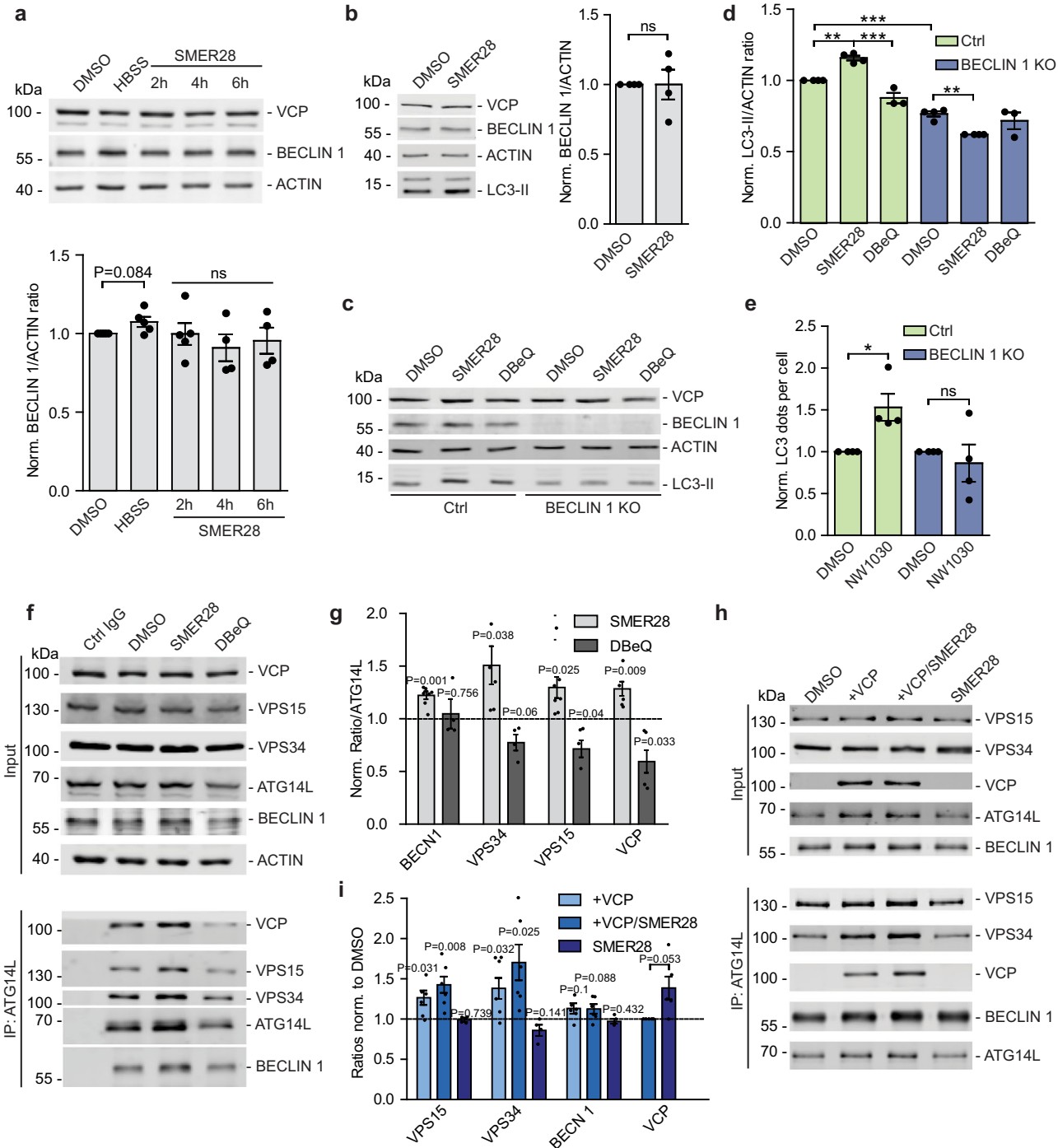

**Fig. 5 SMER28 enhances VCP stimulation of PI3K complex formation to induce autophagy in a Beclin 1-dependent manner. a**, **b** Beclin 1 levels in HeLa cells treated with 20 μM SMER28 for 2, 4, or 6 hours. In the presence of 400 nM BafA1 **b**; $n = 5$. **c**, **d** Beclin 1 knockout cells and control cells were pre-treated with 400 nM BafA1 for 1 h prior to treatment with 20 μM SMER28 or 10 μM DBeQ for 8 h in the presence of BafA1. LC3-II to actin ratios are quantified in **d** $n = 4$; one-way ANOVA: $P < 0.0001$ with post hoc Tukey test, Ctrl SMER28 $P = 0.0027$, DBeQ $P < 0.0001$, Ctrl vs. KO $P < 0.0001$, KO SMER28 $P = 0.0057$. **e** Number of LC3 puncta in Beclin 1 knockout cells and control cells. Cells were pre-treated with 400 nM BafA1 for 1 h prior to treatment with 10 μM NW1030 for 8 h; $n = 4$; control cells $P = 0.0457$, Beclin 1 knockout $P = 0.5854$. **f**, **g** Immunoprecipitation of endogenous ATG14L from HeLa cells treated with 20 μM SMER28 or 10 μM DBeQ for 6 h. Ratios of proteins co-immunoprecipitated with ATG14L in the denoted conditions, normalized to ratios in DMSO control in **g** $n = 7$. **h**, **i** In vitro assembly of PI3K complexes. FLAG-tagged PI3K components were added individually (VPS15 and VPS34 purified and added together) and incubated alone or together with purified VCP and/or 20 μM SMER28, followed by immunoprecipitation of ATG14L. **i** Ratios of proteins co-immunoprecipitated with ATG14L in the denoted conditions, normalized to ratios in DMSO control. Quantification of VCP levels shows VCP/ATG14L ratios with and without the addition of SMER28, normalized to ratio without SMER28; n = 6. See also Supplementary Fig. 5c for differences in PI3K interactions with the addition of VCP with or without additional treatment with SMER28. **f–i** After protein transfer, PVDF membrane was cut and fragments were incubated with appropriate primary antibodies. Data presented as normalized mean ± SEM, *$P < 0.05$, **$P < 0.001$, ***$P < 0.0001$, one sample $t$ test unless stated otherwise; ns not significant, Ctrl control cells. Source data are provided as a Source Data file.

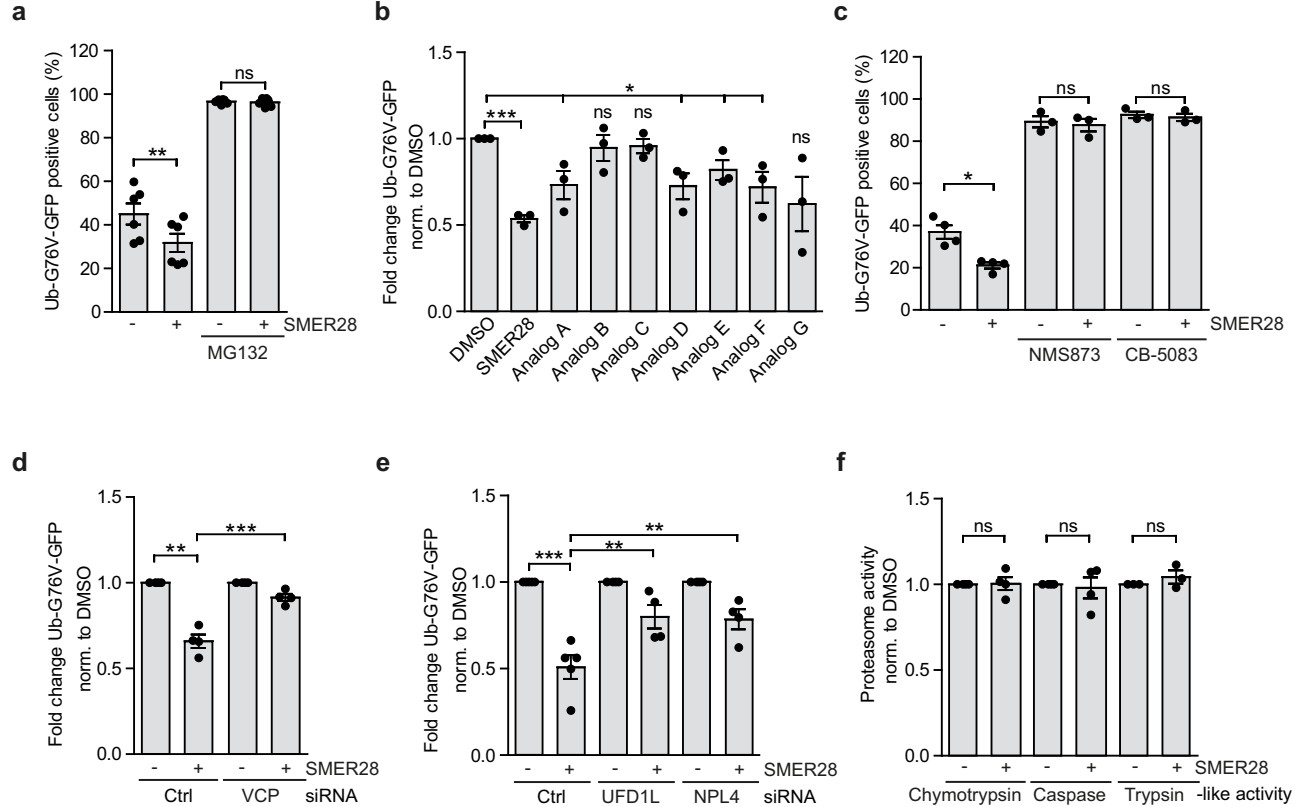

**Fig. 6 VCP activation by SMER28 enhances UPS flux. a** HeLa cells stably expressing Ub-G76V-GFP were treated with 20 μM SMER28 with or without 5 μM MG132 for 6 h, followed by FACS analysis, $n = 6$, paired two-tailed Student's $t$ test; DMSO $P = 0.0006$; MG132 $P = 0.3509$. **b** HeLa cells stably expressing Ub-G76V-GFP were treated with 20 μM SMER28 or its analogs for 6 h, followed by FACS analysis; unpaired two-tailed Student's $t$ test, $n = 3$; SMER28 $P = 0.00002$, A $P = 0.03$, B $P = 0.5516$, C $P = 0.3542$, D $P = 0.0214$, E $P = 0.0337$, F $P = 0.0344$, G $P = 0.748$. **c** HeLa cells stably expressing Ub-G76V-GFP were treated with 20 μM SMER28 with or without 10 μM NMS873 and 5 μM CB-5083 for 6 h, followed by FACS analysis, $n = 4$; paired two-tailed Student's $t$ test; DMSO $P = 0.0089$, NMS873 $P = 0.4895$, CB-5083 $P = 0.0319$. **d, e** HeLa cells stably expressing Ub-G76V-GFP were treated with siRNA against VCP (**d** $n = 4$) and siRNA against UFD1L or NPL4 (**e** $n = 5$). Raw data comparing effect of the knockdown in Supplementary Fig 5h. After 48 h cells were treated with 20 μM SMER28 or DMSO for 6 h, followed by FACS analysis; data normalized to DMSO control across all samples to allow comparison of SMER28 effect sizes within and between control and siRNA treated cells; statistical analysis for **d, e** one-way ANOVA: $P < 0.0001$ with post hoc Tukey test $P < 0.0001$. **f** Proteasome activity. HeLa cells were treated with 20 μM SMER28 or DMSO for 6 h, lysed and chymotrypsin-like, caspase-like and trypsin-like proteasome activities were measured using fluorogenic peptide substrates, $n = 4$; one sample $t$ test. Data in bar graphs presented as normalized mean ± SEM. *$P < 0.05$, **$P < 0.001$, ***$P < 0.0001$; ns not significant. Source data are provided as a Source Data file.

by levels of BiP or phosphorylation of eIF2α (Supplementary Fig. 6f, g). Consistent with our data, the levels of puromycin-induced misfolded proteins were also decreased upon treatment with NW1030 (Fig. 7d, e).

Next, we compared the levels of misfolded proteins upon treatment with SMER28 in cells depleted of VCP or UFD1L/NPL4 by siRNA (Fig. 7f–i). We observed that knockdown of VCP and UFD1L/NPL4 decreased the levels of puromycin-labeled proteins, likely due to a decrease in protein translation caused by prolonged ER stress present in these cells[56]. Most importantly, SMER28-mediated decrease in the levels of misfolded proteins was blocked in cells where VCP and UFD1L/NPL4 were depleted (Fig. 7f–i) or treated with Lactacystin, a potent proteasome inhibitor that causes accumulation of K48-ubiquitinated proteins (Fig. 7j, k), indicating that SMER28 enhances degradation of soluble misfolded protein through the UPS.

Treatment with SMER28 or NW1030 decreased the levels of puromycin-induced misfolded proteins in both control and ATG16L1 knockout (autophagy-null) cells (Fig. 7l, m; Supplementary Fig. 6h, i). The decrease in the levels of misfolded proteins upon SMER28 treatment was still observed in cells where lysosomal degradation was impaired by treatment with Bafilomycin A1

(Supplementary Fig. 6j, k), further supporting our observation that the SMER28-induced clearance of short-lived soluble misfolded proteins is independent of the autophagy-lysosomal pathway.

**SMER28 induces degradation of misfolded and aggregate-prone proteins in UPS and autophagy-dependent manner.** Prolonged treatment with puromycin-induced formation of foci co-stained with an antibody recognizing polyubiquitinated proteins, which were not detectable in cells co-treated with cycloheximide, a translation inhibitor, confirming the specificity of the assay (Supplementary Fig. 7a). Treatment with SMER28 prevented the formation of puromycin-induced ubiquitin-positive inclusions and this effect was dependent on the VCP ATPase activity as it was not observed in cells co-treated with VCP inhibitors NMS873 or CB-5083 (Fig. 8a, b). Furthermore, SMER28 significantly accelerated the clearance of puromycin-induced inclusions after puromycin washout (Supplementary Fig. 7b), suggesting that an increase in VCP D1 ATPase activity also affects the well-established VCP function in inclusion clearance[57]. SMER28-mediated decrease in foci formation was also blocked in cells depleted of UFD1L and NPL4 (Fig. 8c;

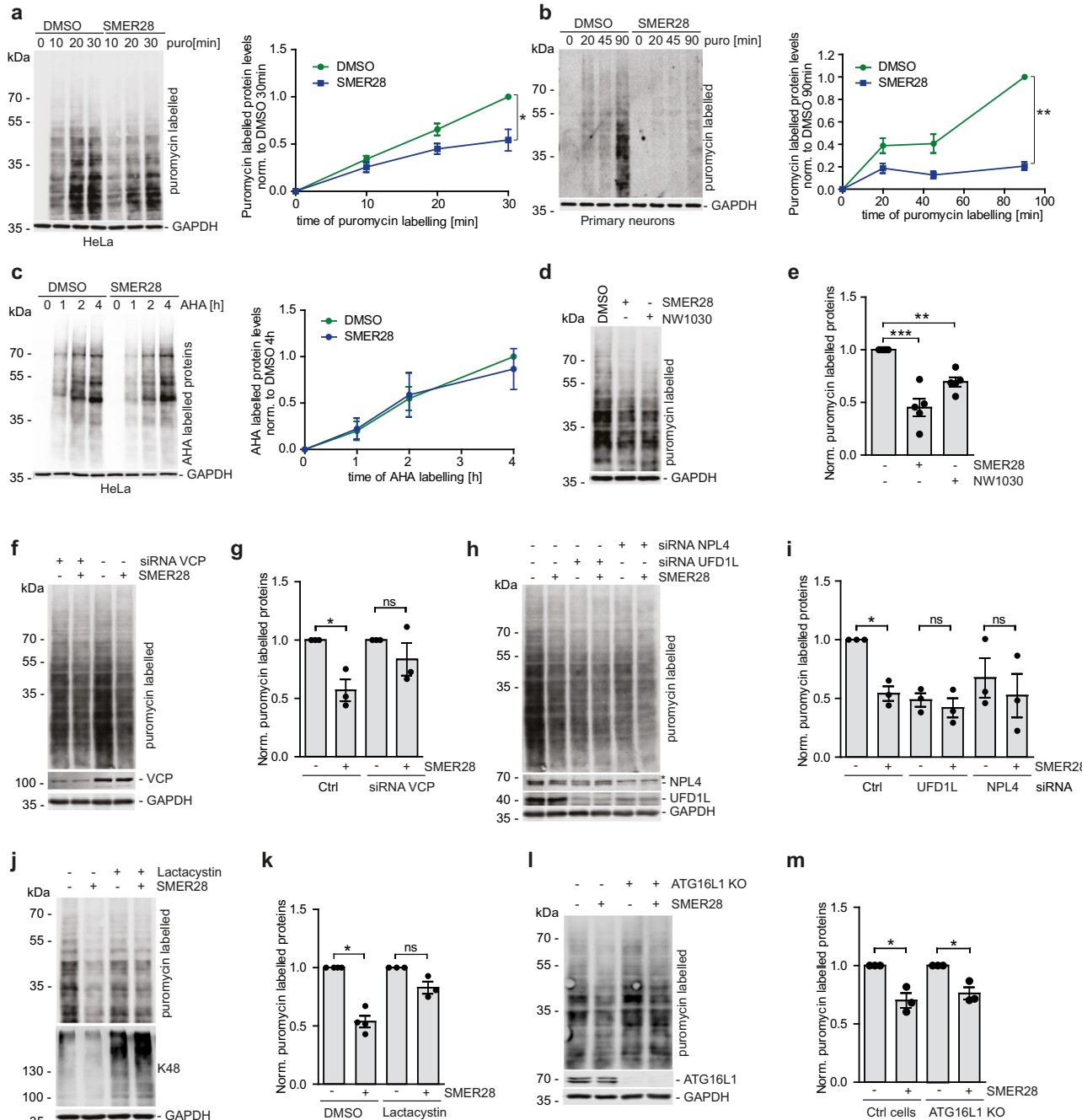

**Fig. 7 SMER28 enhances degradation of short-lived misfolded proteins by UPS in VCP and UFD1L/NPL4 dependent manner. a, b** HeLa cells (**a**) or primary cortical neurons (**b**) were pre-treated with 20 μM SMER28 or DMSO for 4 h, followed by treatment with puromycin for indicated time points, n = 3; (**a**) P = 0.0478, (**b**) P = 0.0022. **c** HeLa cells were pre-treated for 1 h with 20 μM SMER28 or DMSO in a medium deprived of methionine, followed by the addition of methionine analog L-azidohomoalanine (AHA) for indicated time points. Cells were lysed and analyzed by Western blotting using streptavidin antibody, n = 3. **d, e** HeLa cells were pre-treated with 20 μM SMER28, 10 μM NW1030, or DMSO for 4 h, followed by treatment with puromycin for 15 min, n = 5; quantification in (**e** one-way ANOVA: P < 0.0001 with post hoc Tukey test, SMER28 P < 0.0001, NW1030 P = 0.0213). **f–i** HeLa cells was treated with siRNA against VCP (**f, g**) and UFD1L or NPL4 (**h, i**). After 48 h cells were pre-treated with 20 μM SMER28 or DMSO for 4 h, followed by treatment with puromycin for 15 min, n = 3; quantification in (**g** one-sample t test, Ctrl P = 0.0442) and (**I** SMER28 P = 0.0175). **j, k** HeLa cells were pre-treated with 20 μM SMER28 or DMSO for 3 h, followed by 1 h pre-treatment with 2 μM Lactacystin and 15 min with puromycin, n = 3; quantification in (**k**, one sample t test, DMSO P = 0.0026). **l, m** ATG16L1 knockout, and control cells were pre-treated with 20 μM SMER28 or DMSO for 4 h, followed by treatment with puromycin for 15 min, n = 3; quantification in (**m** one sample t test, control P = 0.0415, ATG16 KO P = 0.046). (**a, b, d, f, h, j, l**) Cells were lysed and analyzed by Western blotting. Puromycin-labeled proteins were detected using anti-puromycin antibody. Bar graphs data presented as normalized mean ± SEM. *P < 0.05, **P < 0.001, ***P < 0.0001, paired two-tailed Student's t test unless stated otherwise; *, unspecific band; ns not significant. Source data are provided as a Source Data file.

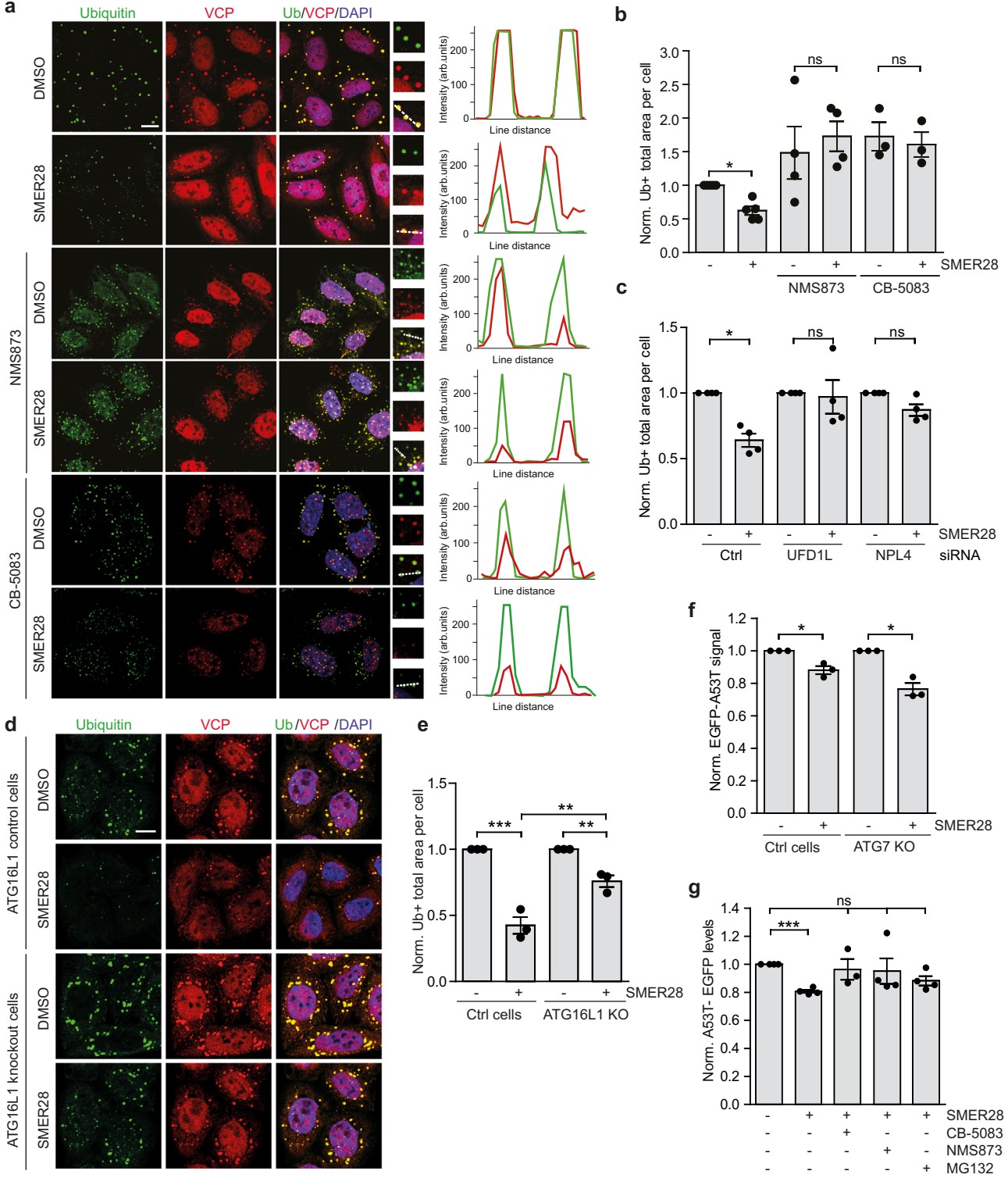

Supplementary Fig. 7c). Treatment with VCP ATPase inhibitors and siRNA-mediated knockdown of UFD1L and NPL4 decreased the colocalization between VCP and ubiquitin-positive inclusions (Fig. 8a; Supplementary Fig. 7c–e), so it is possible that VCP ATPase activity may also influence its ability to interact with certain substrates.

Next, we analyzed the ubiquitin-positive aggregates in control and ATG16L1-null cells treated with puromycin. In agreement with previously published data[57] cells lacking functional

autophagy accumulated puromycin-induced inclusions (Fig. 8d, e). Although treatment with SMER28 could still decrease the levels of ubiquitin-positive inclusions in autophagy-null cells, it was significantly less prominent compared to control cells (Fig. 8d, e), suggesting that SMER28 treatment enhances the clearance of misfolded proteins both through autophagy (e.g. puromycin-dependent inclusions in cells) and UPS-dependent pathways (e.g. gel-soluble puromycylated proteins) simultaneously.

**Fig. 8 SMER28 induces degradation of misfolded and aggregate-prone proteins in UPS and autophagy-dependent manner. a, b** HeLa cells were pre-treated with 20 μM SMER28 with or without 10 μM NMS873 or 5 μM CB-5083 for 1 h, followed by addition of puromycin for 4 h. Cells were fixed and stained for VCP puncta and ubiquitin-positive structures, quantification of the total area of ubiquitin-positive foci in (**b** paired two-tailed Student's *t* test, DMSO *P* = 0.0039). Line scans indicate the degree of colocalisation between VCP (red) and Ubiquitin (green) in lines drawn within the magnified images. The intensity profiles are presented as arbitrary units (arb. units); n = 5; Manders' Colocalisation Coefficient analysis in Supplementary Fig. 7d. **c** HeLa cells were treated with siRNA against UFD1L or NPL4 and after 48 h were pre-treated with 20 μM SMER28 or DMSO for 1 h followed by treatment with puromycin for 4 h, representative images in Supplementary Fig. 7c and Manders' Colocalisation Coefficient analysis in Supplementary Fig. 7e, n = 4; one sample *t* test, Ctrl *P* = 0.0058. **d, e** ATG16L1 knockout and control cells were pre-treated with 20 μM SMER28 or DMSO for 1 h, followed by addition of puromycin for 3 h, statistical analysis in (**e** one-way ANOVA: *P* < 0.0001 with post hoc Tukey test, Ctrl SMER28 *P* < 0.0001, Ctrl vs. KO SMER28 *P* = 0.0012, KO SMER28 *P* = 0.0094), n = 3. **f** HeLa cells stably expressing A53T-SNCA-EGFP in wild-type or ATG7 knockout cells were treated with 20 μM SMER28 for 24 h, followed by FACS analysis; n = 3; one sample *t* test, WT *P* = 0.041, ATG7KO *P* = 0.0244. **g** ATG7 knockout HeLa cells stably expressing A53T-SNCA-EGFP were treated with 0.5 μM CB-5083, 1 μM NMS873, or 1 μM MG132 combined with 20 μM SMER28 for 24 h, followed by FACS analysis; n = 4; Kruskal–Wallis test: *P* = 0.0199 with post hoc Dunn's Multiple comparison test, SMER28 *P* = 0.0129. Bar graphs data presented as normalized mean ± SEM. \**P* < 0.05, \*\**P* < 0.001, \*\*\**P* < 0.0001; scale bar = 10 μm; ns not significant, Ctrl control cells. Source data are provided as a Source Data file.

As the SMER28-mediated lowering of levels of puromycin-labeled proteins that can enter gels ("soluble") can occur in autophagy-null cells and is proteasome-dependent (Fig. 7l, m), we hypothesized that the proteasome route may be more important for SMER28-dependent clearance of soluble disease-causing proteins. To test this further, we treated wild-type and ATG7 knockout (autophagy-null) HeLa cells stably expressing α-synuclein-A53T mutant with SMER28 and observed a similar decrease in the levels of soluble forms of this protein in both cell lines (Fig. 8f; Supplementary Fig. 7f). We confirmed this using cycloheximide chase experiment (Supplementary Fig. 7g). Indeed, we confirmed that A53Tα-synuclein levels are increased in cells where proteasome or VCP activity were inhibited (Supplementary Fig. 7h) and this inhibition prevented a SMER28-mediated decrease in the levels of α-synuclein (Fig. 8g) indicating that treatment with SMER28 enhances clearance of soluble mutant α-synuclein in a UPS and VCP-dependent manner.

## Discussion

Here, we have identified VCP/p97 as a molecular target of SMER28. SMER28 acts by binding VCP at the cleft formed between VCP's substrate binding domain and ATPase domain 1 to increase the ATPase activity of D1 domain. The findings that SMER28 acts similarly to the recently published VCP activator NW1030 reinforces our hypothesis that its protein clearing activities are mechanistically dependent on its binding to VCP.

Both SMER28 and NW1030 increase VCP D1 ATPase activity. While the major ATPase-driven functions of VCP are attributed to the activity of the D2 domain[58], ATP binding to the D1 domain is important for hexamer formation[40] and is tightly linked with the conformation of the N-terminal domain, which regulates substrate and cofactor binding[7,59]. The increase in D1 ATPase activity of VCP correlates with enhanced autophagy initiation by strengthening its interactions with PI3K complex I, increasing the formation of this complex and stimulating PI(3)P production. These effects of SMER28 depend on VCP ATPase activity and are consistent with our recent observation that VCP inhibition decreases PI3K complex I assembly, PI(3)P formation, and autophagosome biogenesis[15] and suggests that enhancing PI3K complex assembly should be considered as a promising therapeutic strategy to stimulate autophagy, for example in neurodegenerative diseases.

Our study revealed that in addition to having a stimulatory effect on autophagy, SMER28 binding to VCP and the correlated increase in its D1 ATPase activity also increased the flux through the UPS, preferentially for non-native proteins. One of the key functions of VCP is to extract and unfold polyubiquitinated substrates and by this prepare them for subsequent degradation by proteasome[60]. Substrates initially bind to UFD1L/NPL4 and then move through the pore of two ATPase rings which causes their unfolding. Biochemical studies have demonstrated that substrate translocation is driven by ATP hydrolysis in the D2 domain which provides a pulling force to move the polypeptide through VCP central pore[52,61–63]. Disease-causing missense mutations in VCP, most of which lie at the interface between the N-terminal and D1 domains, exhibit increased D2 ATPase activity[32,38,64,65], associated with the faster rate of substrate unfolding, thereby suggesting a gain-of-function model[39]. Importantly, SMER28 binding to VCP does not affect the D2 domain activity.

Less is known about the role of ATP hydrolysis driven by D1 domain and how a selective increase in the D1 ATPase activity could affect substrate recognition and unfolding. Here we show that a SMER28-mediated increase in D1 domain activity is correlated with enhanced preferential degradation of misfolded and aggregate-prone proteins, rather than all proteins. We showed that these effects depend on VCP ATPase activity and the presence of UFD1L/NPL4 complex. Recent studies have demonstrated that a subset of the cellular proteome, favoring misfolded and aggregate-prone proteins, is decorated with branched K11/K48 ubiquitin chains, which enhance their targeting for proteasomal degradation[66,67]. VCP was shown to bind K11/K48 branched chains with higher affinity than homotypic K48 chains[61,67], although the mechanism is not understood. It is plausible that a selective increase in ATP hydrolysis by D1 domain, using for example small molecules like SMER28 or NW1030, could induce certain changes in cofactor binding that favor recognition of misfolded substrates decorated with branched K11/K48 ubiquitin chains. We cannot easily address whether the protein clearance effects of SMER28 are specifically or exclusively due to enhanced D1 ATPase activity. For example, SMER28 may have allosteric effects on VCP which result in protein clearance enhancement and the modest increase in D1 ATPase activity may be an epiphenomenon.

The therapeutic potential of SMER28 to combat neurodegenerative diseases has previously been suggested in cultured cells and in animal models of Huntington's, Parkinson's, and Alzheimer's disease[19–23,46]. Here we show that SMER28 and NW1030 treatment induces degradation of polyQ-expanded mutant proteins in mouse striatal cells and in fibroblasts derived from Huntington's disease and spinocerebellar ataxia type 3 patients. Importantly, SMER28 does not appear to reduce the levels of wild-type huntingtin or Ataxin 3, suggesting that VCP binding compounds like SMER28 enable preferential clearance of the mutant/misfolded species and preserve the levels of the wild-type/normally-folded counterparts. This may be an important advantage in the context of neurodegenerative diseases. Furthermore, SMER28 enables the clearance of the misfolded species

by both the proteasome and autophagy routes—our data suggest that the proteasome may be degrading the monomeric substrates, while autophagy may handle the oligomeric species that are inaccessible to the proteasome. Although the pharmacokinetics of SMER28 have not been reported, this compound appears to be well tolerated by animals and appears to induce autophagy in rat brains after intraperitoneal injections, thereby showing the ability of the molecule to cross the blood-brain-barrier. Altogether, these characteristics of SMER28 raise its potential utility, or at least its promise as a start for developing improved molecules for therapeutic purposes[22,68].

## Methods

**Cell lines.** Human cervical epithelium HeLa (ATCC; #CCL-2; CVCL_0030), human embryonic kidney cell line HEK293 (ECACC; #85120602), and striatal neuronal cell lines derived from wild-type HTT Q7/Q7 and homozygous HTT Q111/Q111 knock-in mice (Coriell Institute #CH00097 and #CH00095, respectively) were cultured in Dulbecco's modified Eagle's medium (DMEM) (4.5 mg/L of glucose; Sigma) supplemented with 10% v/v FBS (Sigma), 2 mM L-glutamine (Sigma) and 100 U ml$^{-1}$ penicillin/streptomycin (Sigma). Human embryonic suspension cells, Expi293F (Gibco; #A14527), were grown in Expi293 Expression Medium (Gibco). SH-SY5Y cells (ECACC, #94030304) were grown in DMEM/F-12 (Gibco) supplemented with 10% v/v FBS (Sigma), 2 mM L- glutamine (Sigma) and 100 U ml$^{-1}$ penicillin/streptomycin (Sigma). For primary cortical neurons, the cortex was dissected from embryonic day 16.5 C57BL/6 mice cross and cultured in Neurobasal®-A MediumMinus Phenol Red (Life technologies), containing 1× B-27®Serum-Free Supplement (50×), Liquid (Life Technologies), 2 mM L-glutamine (Sigma) and 100 U ml$^{-1}$ penicillin/streptomycin (Sigma). Primary fibroblasts from 2 unaffected controls (Ctrl, Coriell Institute #GM04711; #GM04729), 4 Huntington's disease patients (HD, Coriell Institute #GM04476, #GM21756, #GM21757; polyQ17/80 #HD30501 was a kind gift from Ferdinando Squitieri, Huntington, and Rare Diseases Unit, Fondazione IRCCS Casa Sollievo della Sofferenza Research Hospital, Italy) and spinocerebellar ataxia type 3 patient (SCA3; Coriell Institute #GM06153) were grown at 37 °C in GlutaMAX media (Gibco) supplemented with 20% v/v FBS, 100 U ml$^{-1}$ penicillin/streptomycin, MEM non-essential amino acid solution (Sigma). Ub-G76V-GFP-expressing stable HeLa cell line was described previously[69]. HeLa TALEN BECLIN 1 knockout cell line was kindly provided by Wensheng Wei, Peking University, Beijing[70]. HeLa CRISPR/Cas9 ATG16L1 knockout cell line was generated using a double-nicking strategy with paired guide RNAs and was described previously[71]. Method for primary cortical neurons isolation was described previously[5].

All the cell lines were maintained at 37 °C (except striatal neuronal cell lines which were grown at 33 °C) and 5% CO$_2$ and were regularly tested for mycoplasma contamination. All cell lines were authenticated by the provider company and/or by Western blot analysis of specific proteins. For starvation experiments, cells were washed three times in starvation media (Hank's balanced salt solution (HBSS, Invitrogen) or Earle's balanced salt solution (EBSS, Sigma) and incubated for 2–4 h at 37 °C. For BONCAT method cells were grown in DMEM without methionine and cysteine (Gibco, #21013024) supplemented with 10% FBS (Sigma), 2 mM L-glutamine (Sigma), and 100 U ml$^{-1}$ penicillin/streptomycin (Sigma).

**Antibodies and reagents.** The following primary antibodies have been used in this work: mouse-anti-Flag M2 (Cat# F1804, RRID:AB_262044, 1:1000), and rabbit anti-Actin (Cat# A2066, RRID AB_476693; 1:1000) from Sigma-Aldrich; rabbit anti-VCP (Cat# ab109240, RRID:AB_10862588; 1:2000 for WB; 1:400 for IF), rabbit anti-LC3B (Cat# ab51520, RRID:AB_881429; 1:400 for IF), rabbit anti-GFP (Cat# ab6556, RRID:AB_305564; 1:1000), mouse-anti-GFP (Cat# ab1218, RRID: AB_298911; 1:1000), rabbit anti-VPS15 (Cat# ab128903, RRID: AB_11141464; 1:1000), rabbit anti-VPS34 (Cat# ab227861, RRID: AB_2827796; 1:1000), rabbit anti-BiP (Cat# ab21685, RRID: AB_2119834; 1:1000), mouse-anti-GAPDH (Cat# ab8245,RRID: AB_2107448; 1:1000), rabbit anti-UFD1L (Cat# ab96648, RRID: AB_10678868; 1:1000), rabbit anti-NPL4 (Cat# ab101226, RRID:AB_10862595; 1:500), rabbit anti-CALNEXIN (Cat# ab10286, RRID:AB_2069009; 1:2000), rabbit anti-ATG7 (Cat# ab133528, RRID:AB_2532126; 1:1000), mouse-anti-WIPI2 (Cat# ab105459, RRID:AB_10860881; 1:400) from Abcam; rabbit anti-LC3B (Cat# NB100-2220, RRID: AB_10003146; 1:1000), rabbit anti-TEX264 (Cat# NBP1-89866, RRID:AB_11009420; 1:1000) from Novus Biologicals; rabbit anti-BECLIN 1 (Cat# 3738, RRID: AB_490837; 1:1000), rabbit anti-K48-linkage polyubiquitin (Cat# 8081, RRID:AB_10859893; 1:1000), rabbit anti-phospho-eIF2alpha (Ser51) (Cat# 9721, RRID:AB_330951; 1:1000), rabbit anti-eIF2aplha (Cat# 9722, RRID:AB_2230924, 1:1000), rabbit anti-ATG16L1 (Cat# 8089, RRID:AB_10950320; 1:1000 for WB, 1:400 for IF), rabbit anti-Akt (Cat# 9272, RRID:AB_329827, 1:1000), rabbit anti-phospho-Akt-Ser473 (Cat# 4060, RRID:AB_2315049, 1:1000), rabbit anti-phospho-Akt-Thr308 (Cat# 9275, RRID:AB_329828, 1:1000) from Cell Signaling; rabbit anti-ATG14L (Cat# PD026, RRID: AB_1953054; 1:1000), mouse-anti-ATG14L (Cat# M184-3, RRID: AB_10897331; 1:1000), rabbit anti-p62 (Cat# PM045, RRID:AB_1279301; 1:2000) from MBL; and mouse-anti-SCD1 from ATS

bio (Cat# AB-259, RRID: AB_888013; 1:1000); mouse-anti-puromycin (Cat# MABE343, RRID:AB_2566826; 1:1000), mouse-anti-Huntingtin (Cat# MAB2166, RRID:AB_2123255; 1:1000), mouse-anti-Polyglutamine-Expansion (Cat# MAB1574, RRID:AB_94263; 1:1000), mouse-anti-Ataxin 3 (Cat# MAB5360, RRID:AB_2129339; 1:1000) from Millipore; mouse-anti- Mono- and poly-ubiquitinylated conjugates (FK2) (Cat# BML-PW8810, RRID:AB_10541840; 1:400 for IF) from Enzo LifeSciences. All the primary antibodies were used with overnight incubation at 4 °C, unless otherwise stated, and the secondary antibodies are used at a concentration of 1:2000 and incubated for 1 h at room temperature. For immunoprecipitation experiments, light-chain specific secondary antibodies were used at a 1:1000 dilution for 1 h at room temperature.

Reagents used include: BafA1 (Enzo LifeSciences, #BML-CM110), Torin 1 (Tocris, #4247), Rapamycin (LC Laboratories, #R-5000, Cycloheximide (Sigma, #C7698), VPS34-IN1 (Seleckchem, #S7980), DbeQ (Tocris, #4417), CB-5083 (Seleckchem, #S8101), NMS873 (Tocris, #6180/5), MG132 (Sigma #C2211), Lactacystin (Stratech Scientific #A2583), SMER28 (Tocris, #4297), SMER28 structural analogs and NW1030 (synthesized by AstraZeneca, see part below).

**DNA constructs and siRNA.** The following DNA constructs were used in this study: p3XFLAG-Beclin 1 (#24388), pStrep-Strep-FLAG-VPS15 (#99326), pStrep-Strep-FLAG-VPS34 (#99327), EGFP-a-synuclein-A53T (#40823), pET41b + _Ufd1-HIS (#117107), pET41 + b_Npl4-HIS (#117108). VCP(wt)-EGFP (#23971) from Addgene; pEGFP-N1 (#6085-1) from Clontech. p3XFLAG-ATG14L was a gift from Zhenyu Yue (The Friedman Brain Institute, Icahn School of Medicine at Mount Sinai, New York, USA). pGEX-VCP-GST and pGEX-VCP-R155H-GST were kindly shared by Rolf Schröder and Cristoph Clemen (University Hospital Erlangen, Erlangen, Germany). Pre-designed siRNAs (ON-TARGETplus SMARTpool) from GE Healthcare Dharmacon were used: control non-targeting siRNA (#D-001810-10), VCP siRNA (#L-008727-00-0005), NPL4 siRNA (#L-020796-01-0005), UFD1L siRNA (#L-017918-00-0005).

**Mutagenesis to produce VCP Walker B mutants.** pGEX-VCP-GST vector was used as a template for site-directed mutagenesis to produce VCP ATPase mutants E305Q and E578Q. E305Q mutant was produced using Quikchange Site-Directed mutagenesis kit (Agilent) using following primers: VCP_E305Q_F: 5′-CATCA TCTTCATTGATCAGGTCTAGATGCCATCG-3′; VCP_E305Q_R: 5′-CGATGGCA TCTAGCTGATCAATGAAGATGATG-3′; E578Q mutant was produced using Q5 mutagenesis kit (New England Biolabs, #E0554S) with the following primers: SMH31: 5′-ATTCTTTGATcagCTGGATTCGATTGCCAAGG-3′; SMH32: 5′-AGCACACAGGGGGCAGCT-3′. Mutagenesis reactions were performed according to the manufacturer's instructions. All constructs were verified by sequencing.

**SRAI-LC3B and EGFP-α-synuclein-A53T stable Cell Line Generation.** The SRAI sequence was subcloned from pcDNA3/SRAI (a kind gift from Atsushi Miyawaki, RIKEN Center for Brain Science). The SRAI reporter was cloned in frame into the 5′-end of hLC3B-pcDNA3.1 (previously described in Jahreiss et al. 2008) using KpnI and BamHI restriction sites to generate a pcDNA3.1-SRAI-hLC3B plasmid. To generate stable cell lines, pcDNA3.1-SRAI-hLC3B was linearized via digestion with BglII and transfected into HeLa cells. Starting from 48 h post transfection, stably transfected cells were cultured for 10 days in media supplemented with G418 (Gibco #11811031). To generate single cell clones, SRAI-LC3B expressing cells that emitted both blue and yellow fluorescence were selected by FACS and sorted into 96-well plates containing one cell per well. These cells were subsequently expanded to generate monoclonal lines. EGFP-α-synuclein-A53T (Addgene Plasmid #40823) was cloned into the pIRES2 DsRed-Express2 vector (Clontech #632540) using the NheI and SacII. A linearized subcloned vector was used to transiently transfect HeLa cells, followed by selection with G418. Cells expressing medium fluorescence for both green and red wavelengths were sorted using a BD Influx Cell sorter (BD Biosciences), expanded, and clones were selected based on their response to modulators of autophagy and expression of green and red fluorescence. EGFP-α-synuclein-A53T ATG7 knockout cell line was generated using CRISPR/Cas9 method using the following gRNA sequence: 5′-CACCGGAACTTGTTGAGGAG TACAGT-3′; 5′-TAAAACTGTACTCCTCAACAAGTTCC-3′.

**Transfection.** Trans IT-2020 reagent (Mirus, #MIR5400) was used for DNA transfection, while Lipofectamine 2000 (Invitrogen, #11668019) was used for siRNA transfections, according to the manufacturer's instructions. For protein production in suspension Expi293F cells, transfection was performed with Polyethylenimine (PEI), at a ratio of 3:1 PEI:DNA. After transfection, cells were maintained in full medium. For knockdown experiments, cells were transfected with a single round of 50 nM siRNA in Opti-MEM reduced serum media (Gibco, #31985070). Cells were split 24–48 h after transfection, and harvested 2–3 days post-transfection.

**Western blot analysis.** For denaturing gel conditions, cells were lysed in Laemmli sample buffer and boiled for 10 min at 100 °C, separated by SDS-PAGE, transferred to PVDF membranes, and subjected to Western blot analysis and visualized direct infrared fluorescence detection on an Odyssey Infrared Imaging System (LICOR). In some experiments, after protein transfer, PVDF membrane was cut into fragments to allow for incubation with different primary antibodies. For native gel

conditions, cells were lysed in digitonin-containing buffer (1% digitonin, 50 mM Tris pH 7.4, 2 mM ATP, protease inhibition cocktail) by passing cells 10 times through a 30 G syringe. After a clarifying spin (20,000 × g, 10 min, 4 °C), protein concentrations were determined by Bradford assays with bovine serum albumin as standard. Equal amounts were mixed with Native-PAGE™ Sample Buffer (4×) and G-250 Sample Additive and separated using Blue-Native-PAGE (Thermo Scientific). Densitometric analysis on the immunoblots was performed using IMAGE STUDIO Lite software, which enables quantitative analysis of blotting signals.

**Immunofluorescence.** The staining of PI(3)P was performed as described previously[15,72]. Briefly, cells were fixed in 2% w/v paraformaldehyde, permeabilized with 20 µM digitonin, and blocked with 5% v/v FBS. Mouse-anti-PI(3)P antibody (1:300; 1 h at room temperature) (Echelon Biosciences Cat# Z-P003, RRI-D:AB_427221) and secondary antibody (1:400; 30 min at room temperature) (goat-anti-mouse Alexa Fluor 555; ThermoFisher, #A21147) were applied, followed by post-fixation in 2% paraformaldehyde, washing and mounting on microscope slides with ProLong Gold Antifade Mountant with DAPI (ThermoFisher). For imaging of LC3 puncta, cells were fixed in ice-cold methanol for 5 min, blocked in 1% BSA at room temperature for 1 h, then incubated with rabbit anti-LC3B (Abcam, #ab192890) overnight, and with secondary goat-anti-rabbit Alexa Fluor 594 (ThermoFisher, #A11012) for 1 h at room temperature. For imaging of WIPI2, ATG16, Ubiquitin, and VCP, cells were fixed in paraformaldehyde 4% for 10 min and permeabilized with 0.1–0.2% Triton X-100 for 5–10 min, then incubated with indicated primary antibodies for overnight in 4 °C, and with secondary Alexa Fluor antibodies for 1 h at room temperature. Imaging for puromycin-labeled proteins was performed as previously described[73]. Briefly, O-propargyl-puromycin (Jena Bioscience, #NU-931-05) labeled cells were fixed in ice-cold methanol for 2 min at −20 °C, washed with PBS, and permeabilized with 0.2% Triton X-100. Cells were stained for 30 min in 100 mM Tris pH 8.5, 0.5 mM CuSO4, 20 µM Alexa Fluor 594-azide (ThermoFisher #A10270), and 50 mM ascorbic acid, followed by wash. Aggregates in fibroblasts were detected using PROTEOSTAT® Aggresome detection kit (Enzo LifeSciences, # ENZ-51035) following the manufacturer's protocol. Briefly, cells were fixed in 4% PFA for 15 min at room temperature and permeabilized with 0.5% Triton X-100, 30 mM EDTA for 30 min on ice. Cells were stained with proteostat solution (1:500) for 30 min at room temperature, followed by 15 min wash in PBS. Coverslips were mounted with ProLong Gold Antifade Reagent (ThermoFisher). Imaging was conducted with LSM710 or LSM880 Zeiss confocal with ×63 oil-immersion lens.

**Protein synthesis and puromycin-induced foci analysis.** The levels of newly synthetized proteins were measured using either BONCAT[54] or SUnSET methods[74]. For BONCAT method cells were grown in DMEM without methionine for 1 h followed by the addition of the methionine analog L-azidohomoalanine (AHA) (ThermoFisher, #C10102) for 1–4 h. Cells were lysed in RIPA buffer (150 mM NaCl, 1% v/v Triton X-100, 0.5% sodium deoxycholate, 0.1% w/v SDS, 50 mM Tris 8.0, protease inhibition cocktail) and proteins labeled with azide (AHA) were detected using Click-iT™ Protein Reaction Buffer Kit (ThermoFisher, #C10276) following manufacturer's protocol. AHA-labeled proteins were visualized by Western blotting and subsequent detection with streptavidin-Alexa Fluor 488 (ThermoFisher, #S11223). For analysis of puromycin-labeled protein levels with the SUnSET method cells were incubated with 10 µg/mL of puromycin (ThermoFisher, #A1113803) for 5–90 min in full growth media, followed by lysis in 4 M Urea sample buffer. Puromycin-labeled proteins were visualized by Western blotting and subsequent detection with an anti-puromycin antibody (Millipore, #MABE343). For analysis of puromycin-induced foci, cells were incubated with O-propargyl-puromycin (Jena Bioscience #NU-931-05) for 2 h prior to fixation. For analysis of ubiquitin-positive puromycin-induced foci formation cells were incubated with 5 µg/mL of puromycin for 3–4 h in full growth media, followed by fixation in 4% w/v PFA and staining with FK2 antibody (Millipore, #04-263).

**Measurement of proteasome activity with fluorogenic peptide substrates.** Hydrolysis of fluorogenic substrates suc-LLVY-AMC (Enzo LifeSciences, #BML-P802), Boc-LRR-AMC (Enzo LifeSciences, #BML-BW8515), and Z-LLE-AMC (Enzo LifeSciences, #BML-ZW9345) were measured to determine the proteolytic activity of the chymotrypsin-like, trypsin-like and caspase-like sites of proteasomes. Cells were resuspended in lysis buffer (50 mM Tris-HCl pH 7.4, 10 mM MgCl2, 10% glycerol, 2 mM ATP, 2 mM PMSF, 1 mM DTT) and shaken with glass beads (Sigma-Aldrich) for 10–20 min at 4 °C. After a clarifying spin (20,000 × g, 15 min, 4 °C), protein concentration was determined by Bradford assay with bovine serum albumin as standard. Activity assays were performed in a final volume of 200 µl of lysis buffer with 100 µg of soluble total protein extracts in a 96-well plate by adding 100 µM peptide substrates. Fluorescence (excitation wavelength 380 nm, emission wavelength 460 nm) was measured every 5 min for 1–2 h at 25 °C using a microplate fluorometer (Tecan).

**FACS analysis of mutant α-synuclein (A53T), Ub-G76V-GFP, and SRAI-LC3B.** HeLa cells stably expressing GFP-tagged mutant α-synuclein (EGFP-A53T) or Ub-G76V-GFP were treated with various compounds for 24 h. Cells were then trypsinized and GFP fluorescence was analyzed using an Attune NxT Flow Cytometer

(ThermoFisher Scientific) using the BL1 (488 530/30) detector. Cells were first gated on forward (FSC-A) and side scatter (SSC-A) for P1 and then for singlets (FSC-A/FSC-H) for P2. 20,000 single cells were recorded for each replicate. GFP + gates were set using normal HeLa cells. HeLa cells stably expressing SRAI-LC3B were treated with various compounds for 24–48 h. Cells were trypsinized and analyzed using an Attune NxT Flow Cytometer (ThermoFisher Scientific) using the VL2 (405 512/25) and BL1 (488 530/30) detectors. The ratio of VL2 to BL1 signals was derived for each cell and the median ratio per condition was used for analysis. The data were analyzed using FlowJo software v10.7.1.

**Protein purification.** Purification of GST-VCP, GST-VCP-E305Q, GST-VCP-E578Q, and GST-VCP-R155H from E. coli was performed as described previously[15]. Purification of UFD1-HIS and NPL4-HIS from E. coli was performed as described previously[75]. Briefly, the expression vector was transformed into bacterial strain Rosetta 2 BL21 (DE3) (Novagen) according to instructions from the supplier. Cells from a 1 L culture were harvested after overnight induction of protein expression with 0.2 mM IPTG at 18 °C for VCP protein or with 0.4 mM IPTG at 16 °C for 16 h for UFD1 and NPL4. The cell pellet was resuspended in lysis buffer (2× PBS, 20 mM MgCl2 for VCP; 200 mM KCl, 50 mM Tris-HCl at pH 8.0, 2.5 mM MgCl2, 1 mM ATP, 5% glycerol, 10 mM Imidazole for UFD1/NPL4) containing protease inhibitors and lysed by incubation with 0.5 mg/mL lysozyme and DNase I (1 U/mL) for 30 min on ice, followed by sonication. Lysates were clarified by ultracentrifugation (100,000 × g for 20 min at 4 °C). For UFD1/NPL4, clarified lysates were mixed in 1:1 ratio and incubated gently rotating for 1 h at 4 °C in order to form heterodimers, followed by incubation with Ni-NTA resin (Qiagen) for 2-3 h at 4 °C. UFD1/NPL4 heterodimers were eluted using 250 mM Imidazole, followed by buffer exchange filtration. For VCP purification, clarified lysates were incubated with 1 mL glutathione sepharose resin (Pierce) for 2 h at 4 °C. Resin was added to the gravity flow column and washed with wash buffer (lysis buffer + 0,1% Triton X-100), followed by 3 × 5 min washes with washing buffer containing 1 mM ATP. At this step, the purified protein was either cross-linked to beads to be used in in vitro binding assays or removed from beads by cleaving GST-tag with PreScission Protease. For crosslinking, beads were washed 3× in 200 mM HEPES (pH 8.5), and then incubated with crosslinking solution (20 mM dimethyl pimelimidate DMP in 200 mM HEPES pH 8.5) at room temperature for 60 min. Crosslinking solution was removed and reaction was stopped by incubating beads with 0.2 M ethanolamine-HCl (pH 8.2) for 60 min. Beads were washed 3× in washing buffer (150 mM NaCl, 200 mM Glycine-HCl pH 2.0) and stored in binding buffer (25 mM HEPES pH 7.25, 200 mM NaCl, 0.01% Triton X-100, and 5% Glycerol, 1 mM DTT) containing 0.05% sodium azide. For removal of GST-tag, beads were washed 5× with PreScission cleavage buffer (50 mM Tris pH 7.0, 150 mM NaCl, 1 mM EDTA, 1 mM DTT, Triton X-100 0.1%), and then resuspended in cleavage buffer. PreScission protease was added (60–80 U/mL final concentration) and lysate was incubated overnight at 4 °C, followed by the collection of supernatant containing purified VCP.

Purification of FLAG-tagged PI3K complexes from HEK293 was performed as previously described[15]. Purification of FLAG-VPS15 and FLAG-VPS34 from suspension cells to be used in in vitro PI3K assembly assay, was performed as previously described[15].

**Immunoprecipitation (IP).** For immunoprecipitation of endogenous proteins, cells from one 55 sqcm dish were lysed in 0.2 mL of IP buffer (20 mM Tris pH 7.4; 2 mM MgCl2; 200 mM NaCl; protease inhibitors) with 0.5% NP-40, cleared by centrifugation, diluted to 1 mL by addition of IP lysis buffer (final 0.1% NP-40). Lysates were pre-cleared for 1 h by incubation with non-targeting IgG control antibody (mouse-anti-HA or rabbit anti-GFP) and beads for 2 h at 4 °C. Input samples were collected, and lysates were then incubated with primary antibodies overnight at 4 °C, followed by the addition of 30 µL of washed beads (50% slurry). Endogenous ATG14L was immunoprecipitated using magnetic Dynabeads Protein A (Invitrogen). Beads were washed 3time with lysis buffer and proteins were eluted in Laemmli sample buffer by boiling and analyzed by western blot. Endogenous immunoprecipitations were detected with light-chain specific antibodies to avoid interference of heavy chain signal.

**In vitro binding assay with VCP-GST.** VCP purified from E. coli and cross-linked to glutathione sepharose beads (Pierce) was used as bait in in vitro binding studies, and binding to individual PI3K components was performed as previously described[15]. For in vitro binding between VCP and UFD1/NPL4 heterodimer, VCP, and UFD1/NPL4 were purified from E.coli and VCP was cross-linked to glutathione sepharose beads (Pierce) to be used as a bait. 500 ng of UFD1/NPL4 complex was incubated in 500 µl of binding buffer (25 mM HEPES pH 7.25, 200 nM NaCl, 0.01% Triton X-100, 1 mM DTT) with VCP-loaded beads for 2 h at 4 °C, followed by wash, elution in Laemmli sample buffer by boiling and analysis by SDS-PAGE and Western blot.

**In vitro PI3K assembly.** In vitro PI3K assembly was performed by incubating individually purified PI3K components with and without the addition of purified VCP and/or addition of 20 µM SMER28 followed by pulldown of assembled PI3K complexes via ATG14L as described previously[15].

**In vitro VCP ATPase assay**. For ATPase activity of VCP 500 ng of recombinant active GST-VCP (SignalChem, #VCP-195H) was incubated in 50 µL of reaction buffer (10 mM HEPES-KOH pH 7.7, 2.5 mM MgCl$_2$, 50 mM KCl, and 1 mM DTT) together with DMSO, 20 µM SMER28 or DBeQ, NMS873, CB-5083. The reaction was started by the addition of 0,1 mM ATP and carried out for 1 h at room temperature. The reaction was stopped by the addition of 100 µL BIOMOL green (Enzo LifeSciences, #BML-AK111-0250), and after 30 min absorbance at 650 nm was determined and the amount of released phosphate was interpolated from a standard curve. For ATPase activity of WT VCP, ATPase mutants and the VCP-R155H mutant, GST-tagged VCP proteins were purified from *E. coli* (GST-tag removed by PreScission protease; Merck, #GE27-0843-01), and used in ATPase assay buffer (20 mM HEPES-KOH pH 7.7, 20 mM MgCl$_2$, 50 mM KCl, and 1 mM DTT). Reaction was started by the addition of 2 mM ATP and carried out at 37 °C for denoted time points, followed by the addition of 100 µL BIOMOL green and absorbance reading. 200–500 ng of purified protein was used per reaction.

**Drug affinity responsive target stability (DARTS) assay**. Performed as previously described[76]. Briefly, HeLa cells were treated with DMSO or 20 µM SMER28/analogs for 1 h, after which growth medium and drugs were washed away and cells were lysed in a mild lysis buffer (50 mM Tris pH 7.4, 200 mM NaCl, 0.5% Triton X-100, 10% Glycerol, 1 mM DTT) with protease inhibitors. Lysates were cleared by centrifugation, and 1:10 volume of 10× TNC buffer (500 mM Tris-HCl pH 8.0, 500 mM NaCl, 100 mM CaCl$_2$) was added. Lysates were split into 2 × 30 µL aliquots, where one was digested by the addition of 0.125 mg pronase (Roche, #10165921001) and the other was left as an undigested control. Digestion was carried out for 35 min at RT and stopped by the addition of sample buffer and boiling. Digestion of the target protein was analyzed by Western blot.

**In vitro kinase assay**. In vitro kinase assay was performed using the Universal Kinase Activity Kit (R&D systems, #EA004) according to the manufacturer's instructions. A substrate mix containing purified FLAG-VPS34 (10 ng) and ATP was mixed with recombinant ULK1 (10 ng) mixed with CD39L2. Recombinant ULK1 showed an increase in phosphorylation of Malachite Green with a phosphatase-coupled approach, in which the nucleotidase, CD39L2 is used to selectively release phosphate from ADP recognized by Malachite Green. The kinase reaction was performed in a 96-well microplate and the reaction was incubated for 10 min at room temperature. Inorganic phosphates were detected with Malachite Green for 20 min and measured using a TECAN Spark microplate reader at 620 nm. The signal of the negative control was subtracted.

**Kinase profiling of SMER28 and analogs**. SMER28 and analogs were tested at 10 µM in a panel of 123 kinases at ThermoFisher Scientific. Follow-up of selected kinases for SMER28 and analogs was also performed at ThermoFisher Scientific. Data are shown in Supplementary Data 2 and 3.

**Synthesis of SMER28 Analogs and a probe for chemoproteomic profiling**
*$^t$Butyl (2-(2-(2-(((6-bromoquinazolin-4-yl)amino)ethoxy)ethoxy)ethyl)carbamate (Analog G)*. Under N$_2$ at 25 °C, a mixture of 6-bromo-4-chloroquinazoline (2.00 g, 8.21 mmol) and tert-butyl (2-(2-(2-aminoethoxy)ethoxy)ethyl)carbamate (2.04 g, 8.21 mmol) in CH$_3$CN (50 mL) was treated with K$_2$CO$_3$ (2.27 g, 16.4 mmol) and stirred at 70 °C for 2 h. The mixture was filtered through *Celite*, washed with CH$_3$CN and evaporated. The residue was purified by flash chromatography (0 to 8% MeOH in CH$_2$Cl$_2$) to afford the desired product as a colorless oil (3.70 g, 99%). $^1$H NMR (400 MHz, DMSO) δ 1.35 (9H, s), 3.03 (2H, q, *J* = 6.0 Hz), 3.35 (2H, t, *J* = 6.1 Hz), 3.50 (2H, dd, *J* = 5.6, 3.1 Hz), 3.54 (2H, dd, *J* = 5.7, 3.2 Hz), 3.62–3.69 (4H, m), 6.75 (1H, t, *J* = 5.8 Hz), 7.62 (1H, d, *J* = 8.8 Hz), 7.89 (1H, dd, *J* = 8.9, 2.2 Hz), 8.45 (1H, t, *J* = 5.6 Hz), 8.48 (1H, s), 8.57 (1H, d, *J* = 2.2 Hz). m/z (ES$^+$), [M + H]$^+$ = 455, 457. HPLC (TFA): 99.0%, t$_R$ = 1.04 min. Other analogs were prepared using standard procedures analogous to that used for Analog G.

*N-(2-(2-(2-aminoethoxy)ethoxy)ethyl)−6-bromoquinazolin-4-amine (amine 3)*. The carbamate above (3.60 g, 7.91 mmol) was treated with HCl in 1,4-dioxane (40 mL, 160 mmol) at 25 °C under N$_2$ and stirred at 25 °C for 2 h. The mixture was evaporated to dryness and the residue was treated with MeOH and evaporated to afford the desired amine salt as a white solid (3.10 g, 100%). $^1$H NMR (400 MHz, DMSO) δ 2.90–2.92 (2H, m), 3.54–3.63 (6H, m), 3.74 (2H, t, *J* = 5.7 Hz), 3.88 (2H, q, *J* = 5.7 Hz), 7.91 (1H, d, *J* = 8.9 Hz), 8.08 (3H, s), 8.20 (1H, dd, *J* = 8.9, 2.1 Hz), 8.95 (1H, s), 9.14 (1H, d, *J* = 2.1 Hz), 10.87 (1H, t, *J* = 5.6 Hz). m/z (ES$^+$), [M + H]$^+$ = 355, 357. HPLC (TFA) 99.3%, t$_R$ = 0.80 min.

**Chemoproteomic profiling of SMER28**
*Bead preparation*. NHS-activated sepharose beads were loaded at 0.2 µmol/mL with primary amine 3 by incubating together overnight on an end-over-end rotator in DMSO with an excess of triethylamine. The beads were capped with an excess ethanolamine and washed with DMSO, then EtOH, and stored in cold EtOH.

*Chemoproteomic affinity enrichment*. HeLa cellular lysate was freshly prepared in lysis buffer (1% NP-40, 50 mM Tris-HCl, pH 7.8, 150 mM NaCl, 0.1% sodium

deoxycholate, 1 mM EDTA, with 1 Pierce protease inhibitor tablet added per 50 mL) on ice. The soluble fraction was isolated and diluted to 3 mg/mL total protein concentration. SMER28 was incubated with 1 mL of lysate at each concentration (8 µM, 80 µM, 800 µM) for 1 h at 4 °C on a rotisserie rotator, then the lysate with the compound was added to beads derivatized with immobilized compound and incubated for 16 h at 4 °C on a rotator. Each pulldown experiment was further processed according to the distinct procedure below.

*Tryptic digestion and iTRAQ-4 reagent labeling (sample preparation)*. Approximately 30 µL beads from the pulldown experiment were subjected to on-bead tryptic digestion and chemical labeling prior to mass spectrometry analysis. Triethyl ammonium bicarbonate, (50 mM TEAB, 45 µL, pH 8) and the reducing agent tris (2-carboxyethyl) phosphine hydrochloride (100 mM, TCEP) were added to the beads to achieve a final concentration of 5 mM. Reduction continued for 20 min at 55 °C, followed by alkylation by S-methyl methanethiosulfonate (MMTS, 10 mM) for 20 min at room temperature. Samples were treated with ProteaseMAX™ surfactant (4 µg) and (1:25) Trypsin-LysC, Promega) and the digestion proceeded for 16 h at 37 °C. The samples were dried (speed vacuum) and reconstituted with TEAB (30 µL of 0.5 M solution). The four iTRAQ labeling reagent tubes (114, 115, 116, 117) were equilibrated to room temperature, spun down, treated with isopropanol (IPA, 50 µL), and mixed. Each sample was treated with the corresponding iTRAQ-4-PLEX reagent: 114 (DMSO), 115 (8 µM SMER28), 116 (80 µM SMER28), 117 (800 µM SMER28), respectively. Samples were incubated for 2 h at room temperature and quenched with HCOOH. Equal amounts of the four iTRAQ-4 labeled samples from the corresponding replicate were combined into two samples and cleaned with solid phase extraction using C18 solid phase extraction (SepPak tC18 100 µg cartridge, product # WAT036820). These labeled peptides were fractionated using a 6 fractions strong cation exchange (SCX and Pierce SCX Mini spin columns). After SCX fractionation, the samples were dried and reconstituted (800 µl of 0.1% HCOOH). Samples were desalted using C18 solid phase extraction, SepPak tC18 100 µg cartridge (Product # WAT036820). Samples were cleaned using the same procedure as previously described (eluted using 4:1 CH$_3$CN/H$_2$O with 0.1% HCOOH), dried (speed vacuum) and reconstituted (15 µL of 3% CH$_3$CN in H$_2$O with 0.1% HCOOH) and analyzed by mass spectrometry. *LC-MS analysis*. All samples were analyzed on a high-resolution mass spectrometer, Q Exactive™ Plus Hybrid Quadrupole-Orbitrap™ Mass Spectrometer (Thermo Scientific), coupled with either an EASY 1000 nLC system. Nano-electrospray ionization was performed by an EASY-Spray™ source. A 2 column, trap and elute configuration was used for analysis, coupling an Acclaim PepMap 100, 75 µm x 2 cm nano viper, C18 3 µm 100 A trap column to a 50 cm Easy-Spray™ PepMap reverse phase C-18 column (ES803, Thermo Scientific) using mobile phases consisting of 100% LC-MS grade H$_2$O with 0.1% HCOOH for mobile phase A and 100% CH$_3$CN with 0.1% HCOOH for mobile phase B. Peptides were eluted using the following gradient: 2–20% of B in 80 min, 20–32% of B in 30 min and 32–95% of B in 1 min, respectively, at a flow rate of 250 nL/min. A 4-uL injection was used for the first and second replicates for the lysate pulldown samples.

The Q Exactive™ Plus Hybrid Quadrupole-Orbitrap™ Mass Spectrometer (Thermo Scientific) was operated in data-dependent mode using a Full MS/ddMS$^2$ Top12 experiment.

*Data analysis*. Proteome Discoverer version 2.1.1.21 was used.RAW file processing, controlling peptide and protein level false discovery rates, and assembling and quantifying proteins from peptides. The data were searched against a manually reviewed Swiss-Prot human database file, containing 20129 entries (download date 2017-01-05). The Sequest HT algorithm was used for analysis with the following tolerances: full tryptic cleavage with a maximum of two missed cleavages, precursor mass accuracy 10 ppm, fragment mass accuracy 20 mDa, static modification of cysteine with Methylthio (45.988 Da) dynamic oxidation of methionine (15.995 Da), static iTRAQ4plex labeling of lysine (144.102 Da) and any N-terminal modification by iTRAQ4plex (144.102 Da). For protein identification, validation was performed at PSM level using 1% false discovery rate (FDR) determined by percolator algorithm based on q-value and rank 1 peptides. For quantitation, unique and razor peptides were considered with a maximum co-isolation of 100% allowed. The data was visualized using TIBCO® Spotfire® Analyst 7.9.2 using median normalized log2-fold changes for proteins that were quantitated in all doses for both replicates.

**Surface plasmon resonance**. All SPR measurements were run on a BIAcore S200 (GE Healthcare) using running buffer; 10 mM HEPES, 150 mM NaCl, 0.05% Tween 20, 5 mM MgCl$_2$, pH 7.4. Full-length VCP (LD Biopharma) was immobilized on EDC/NHS-activated NID500L chip (Xantec) utilizing the 6xHis-tag for pre-concentration of protein on the chip to a final level of 5500 ± 500 RU before addition of 1 M ethanolamine. Concentration series of compounds (*n* = 3) were dispensed using a Digital dispenser (Tecan), normalized to 0.5% DMSO, and injected at 30 µL/min for 1 min. Binding levels were fitted to a Langmuir 1:1 interaction model to extract steady-state affinity (K$_d$).

**Limited proteolysis under native conditions for global analysis of small molecule binding events**. GST-tagged VCP proteins were expressed in *E. coli* and grown at 37 °C. Expression was induced by IPTG at 16 °C overnight. Cells were

harvested by centrifugation, washed three times with 1× PBS, and snap frozen in liquid nitrogen. To prepare samples for LiP analysis, bacteria were lysed in 500 µl of LiP buffer (100 mM HEPES pH 7.5, 150 mM KCl, 1 mM MgCl₂) using a Precellys Evolution tissue homogenizer using Precellys' micro-organism lysing VK01 tubes (Bertin Corp, #P000914-LYSK0-A). The following program was used: 9000 rpm, 6× 30 s, 1 min break, 4 °C. Lysate was spun at 10,000 × g for 10 min and the supernatant was retained for the LiP protocol. Protein amount was determined using a Pierce BCA Protein Assay Kit (ThermoFisher, #23225) according to the manufacturer's instructions.

The E.coli protein lysate was aliquoted to three independent replicates (100 µg per replicate) and incubated at room temperature (RT) with SMER28 (dissolved in DMSO) for 10 min. A 10-concentration dose–response was used (7–10-fold compound dilutions from a high of 2 mM plus two intermediate concentrations of 1 mM and 100 µM and additionally a vehicle control). The intermediate concentrations were added to provide additional data points to better fit dose–response curves during analysis. Proteinase K (1:100 ratio of enzyme to protein) from Tritirachium album (Sigma) was added and samples were incubated for a further 4 min and were then transferred to a 98 °C heat block for 1 min. After 1 min at 98 °C an equal volume of 10% deoxycholate (to a final concentration of 5%) was added to quench proteinase K activity. This mixture was incubated for a further 15 min at 98 °C.

*Proteome preparation in denaturing conditions.* After incubation at 98 °C samples were reduced for 1 hour at 37 °C with 5 mM tris(2-carboxyethyl)phosphine hydrochloride followed by a 30 min incubation at RT in the dark with 20 mM iodoacetamide. Subsequently, samples were digested for 2 hours at 37 °C with lysyl endopeptidase (1:100 enzyme: substrate ratio) in 2 additional volumes of 0.1 M ammonium bicarbonate (final pH of 8). Following this, samples were further digested for 16 hours at 37 °C with trypsin (1:100 enzyme: substrate ratio). Formic acid was added to a final concentration of 1.5% to precipitate the deoxycholate, the samples were centrifuged at 16,000 × g for 10 min and the supernatant was transferred to a new Eppendorf tube. An equal volume of formic acid was added again and the centrifugation step was repeated. Digests were desalted using C18 MacroSpin columns (The Nest Group) following the manufacturer's instructions and after drying resuspended in 1% acetonitrile (ACN) and 0.1% formic acid. Biognosys' iRT kit (Biognosys AG, Schlieren, Switzerland) was added to all samples according to the manufacturer's instructions.

*Mass spectrometric acquisition.* All samples were acquired by DIA (Data Independent Acquisition). Block randomization of samples was performed prior to acquisition. 2 µg of LiP reaction from each sample was separated using an in-house analytical column (75 µm × 60 cm, PicoFrit PicoTip SELF/P Tip 10 µm Emitters (New Objective, Littleton, MA) packed with CSH-C18 beads (1.7 µm; Waters, Millford MA) connected to an Easy-nLC 1200 (Thermo Scientific, Waltham, MA) and recorded on an Orbitrap Exploris 480 mass spectrometer (Thermo Scientific). Peptides were separated by a 1-hour segmented gradient at a flow rate of 250 nl/min with increasing solvent B (0.1% formic acid, 80% ACN) mixed into solvent A (0.1% formic acid, 1% ACN). Solvent B concentration was increased from 1% according to the following gradient: 6% over 1 min and 36 seconds, 8% over 3 min and 12 seconds, 22% over 24 min and 24 seconds, 30% over 10 min and 24 seconds, 32% over 2 min and 24 seconds, 34% over 2 min, 35% over 1 min and 18 seconds, 37% over 1 min and 36 seconds, 41% over 1 min and 36 seconds, 47% over 50 seconds, 59% over 40 seconds and 90% in 10 seconds. This final concentration was held for 7.5 min followed by a rapid decrease to 1% over 10 seconds, which was then held for 4 minutes to finish the gradient. A full scan was acquired between 330 and 1650 m/z at a resolution of 120,000 (20 ms maximal injection time, AGC was set to 300%). A total of 22 DIA segments were acquired at a resolution of 30,000 (54 ms maximal injection time, AGC was set to 1000%). The normalized collision energy was 27% and the first mass was fixed at 250 m/z.

*Mass spectrometric data analysis.* DIA spectra were analyzed with Spectronaut 14 (Biognosys AG)[77] using the direc-DIA default settings with several modifications. First, in the quantification settings the minor (peptide) grouping was adjusted to 'Modified Sequence' and data filtering was set to 'Q value sparse' with global imputing. Second, in the post analysis perspective the differential abundance grouping was set to 'Minor Group' and 'All Ions' were used as the smallest quantitative unit. Digestion enzyme specificity was set to Trypsin/P and semi-specific. Search criteria included carbamidomethylation of cysteine as a fixed modification, as well as oxidation of methionine and acetylation (protein N-terminus) as variable modifications. Up to 2 missed cleavages were allowed. Files were simultaneously searched against the E.coli UniProt fasta database with isoforms (updated 2020-07-01) and a custom human UniProt fasta including exclusively VCP (updated 2020-07-01), as well as the Biognosys' iRT peptides fasta database (uploaded to the public repository). Further, in brief, retention time prediction type was set to dynamic iRT (adapted variable iRT extraction width for varying iRT precision during the gradient) and correction factor for window 1. Mass calibration was set to local mass calibration. The FDR was estimated with the mProphet approach[78] and set to 1% at both the peptide precursor and protein level. Statistical comparisons were performed on the modified peptide level using fragment ions as quantitative input.

*Dose–response analysis and binding site prediction.* Data was first analyzed for differentially regulated peptides between the three concentrations above the estimated IC50 of the compound (2 µM, 20 µM, and 100 µM) and vehicle using Spectronaut's statistical testing performed on the modified peptide sequence level using all (fragment) ions as the smallest quantitative units. This candidate peptide list was filtered based upon q value <0.01 and an absolute log2-fold-change >0.46. Each peptide in this filtered list was then analyzed using an in-house R script typically used to compute LiP scores. Here, the script was truncated so that data was only subjected to dose–response correlation testing (using the "drc" package (https://www.r-project.org)) on all peptides (modified sequence with fragments ions as quantitative units) at every drug concentration to establish a sigmoidal correlation coefficient. Peptides were then ranked based upon the highest dose–response correlations. The necessary output files from Spectronaut are outlined in the docstring at the start of the R script.

To predict the binding site of SMER28 in VCP we used the triangulation approach previously published for LiP-Quant[41] and a previously published structure of VCP (pdb: 5ftk). In brief, the top three peptides by dose–response correlation were identified in PyMOL (The PyMOL Molecular Graphics System, Version 2.0 Schrödinger, LLC.) and the center of mass of all atoms assigned to the aforementioned peptides was calculated and plotted.

All chemicals and compounds for LiP analysis were purchased from Sigma-Aldrich unless specified otherwise. Lysyl endopeptidase was purchased from Wako Pure Chemical Industries. Sequencing grade trypsin was purchased from Promega.

**Image analysis.** Puncta (PI(3)P, LC3) or total area (Ubiquitin, Proteostat staining) analysis was performed in ImageJ, with manual annotation of cell boundaries using ROI and automatic analysis of number of puncta or total area of puncta per cell using particle analysis plugin, using the same cut-off for puncta identification in all conditions. Manders' Colocalisation Coefficient was measured using JACoP plugin in ImageJ. A minimum of 60 cells was examined for each condition and experiments were repeated at least three times. For WIPI2 and ATG16L puncta analysis, Cellprofiler software was used. Cell boundaries were determined based on the fluorescence of the proteins analyzed. Automatic analysis of the number and area of the puncta per cell were obtained using IdentifyPrimeryObjects. Same settings were used for the analysis of the puncta in all conditions. Images were analyzed using ZEN Black Carl Zeiss Microscopy. Western blots images were quantified by densitometry analysis using ImageStudio Lite software.

**Statistical analysis.** Significance levels for comparisons between groups were determined using GraphPad Prism ver 7 and 8 (GraphPad Software) or Excel (Microsoft office). For Western blots, protein levels were normalized to total forms or a housekeeping protein, such as Actin or GAPDH. Error bars shown in the figures represent as standard error of the mean, unless otherwise stated in figure legends. $P$ values of <0.05 were considered statistically significant. Statistical analysis was performed using one-tailed or two-tailed Student's $t$ test or one-way ANOVA followed by appropriate post hoc test for multiple comparisons. Sample sizes were chosen based on extensive experience with the assay performed. Each experiment was repeated at least three times as an independent biological replicate. The experiments were appropriately randomized and blinded when possible. In experiments where we compare multiple distinct treatments to a control at the same time/experiment, we have used an ANOVA or related test. Otherwise, if the perturbations were done at different times, we use $t$ tests and make this clear. For example, in experiments such as Fig. 2c, where we are testing if inhibitors block the effects of SMER28, we have used $t$ tests for two reasons. First, the experiments all use SMER28, thus samples/conditions are not independent. Second, the major part of the experiment is designed to assess if inhibitors block/blunt the increase in ATPase activity caused by SMER28 so we are testing SMER28 does/does not increase ATPase activity in the presence of diverse inhibitors. More information on statistical analysis is given in figure legends and source data file.

**Reporting summary.** Further information on research design is available in the Nature Research Reporting Summary linked to this article.

## Data availability

Authors can confirm that all relevant data are included in the paper and/or its supplementary information files. All mass spectrometry proteomics data are available in the PRoteomics IDEntifications (PRIDE) database via ProteomeXchange. The Limited Proteolysis data used in this study are available in the PRIDE database under accession code PXD027750. The Chemoproteomic profiling of SMER28 data used in this study is available in the PRIDE database under accession code PXD034712. This study used the following databases: UniProt human database file, containing 20129 entries, was accessed and downloaded on 5th January 2017, UniProt fasta database (with isoforms) for E.coli was accessed on 1st July 2020 via the UniProt databases download page (https://www.uniprot.org/downloads). A custom human Uniprot fasta (containing only VCP—Q0IIN5 (Q0IIN5_HUMAN)) was created from the human UniProt fasta database accessed on 1st July 2020 via the UniProt databases download page (https://www.uniprot.org/downloads). The Biognosys iRT peptides fasta database has been made available via

upload to the public repository linked herein (PRIDE identifier PXD027750). Source data are provided with this paper.

## Code availability

The custom R script used to compute dose–response correlations was modified from the script available via GitHub repository (https://zenodo.org/record/6625705#. YqNQ19PMKUk)[79].

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

## Acknowledgements

We thank Wensheng Wei for sharing the Beclin 1 TALEN KO and TALEN Control HeLa cell lines, Zhenyu Yue for sharing the pFLAG-ATG14L construct, Rolf Schröder and Cristoph Clemen for sharing pGEX-VCP-GST and pGEX-VCP-R155H-GST. We thank Henrik Zetterberg at the University of Gothenburg for sharing facilities, and Maria Olsson for guidance in protein purification. We are grateful for funding from the UK Dementia Research Institute (funded by the MRC, Alzheimer's Research UK, and the Alzheimer's Society) (to D.C.R.), The Tau Consortium, Alzheimer's Research UK, an anonymous donation to the Cambridge Center for Parkinson-Plus, AstraZeneca, the Swedish Natural Research Council (V.R.) (to S.M.H; reference 2016–06605) and from the European Molecular Biology Organisation (EMBO long-term fellowships to SMH and LW; ALTF 1024-2016 and ALTF 135-2016, respectively).

## Author contributions

L.W., S.M.H., and D.C.R. conceptualized the project. L.W. and S.M.H. designed and performed most of the experiments with help of A.D., M.F.-E., C.K. A.D. constructed the stable SRAI-LC3B cell line, V.J.B. constructed the stable EGFP-A53T cell line, E.S. constructed the stable EGFP-A53T ATG7 knockout cell line. A.A. performed some of the initial experiments. Chemoproteomic experiments and target identification and SPR-binding studies were performed by T.R., E.W.M., M.P.C., A.G., R.J., and R.W.B. R.W.B. and K.T. contributed to the design and interpretation of SMER28 experiments. N.B., R.B., Y.F., and L.R. conceived and designed the LiP study. N.B. designed and performed the LiP experiments. N.B. and R.B. analyzed the LiP data. L.W., S.M.H., and D.C.R. wrote the manuscript with input from the other authors.

## Competing interests

E.W.M., T.R., M.P.C., A.G., K.T., R.J., and R.W.B. were employees of AstraZeneca when the experiments were performed and are shareholders of AstraZeneca. R.W.B. is currently employed by Cerevance Ltd. T.R. is currently employed by Monte Rosa Therapeutics. M.P.C. is currently employed by LifeMine Therapeutics. M.F.E. is currently employed by GW Research Ltd. N.B., R.B., Y.F., and L.R. are employees of Biognosys A.G. D.C.R. is a consultant for Alladin Healthcare Technologies Ltd., Mindrank AI, Nido Biosciences, Drishti Discoveries, and PAQ Therapeutics. None of the other authors have competing interests.
