## [Peer Review File · Nature Communications]

Compounds activating VCP D1 ATPase enhance both autophagic and proteasomal neurotoxic protein clearanceREVIEWER COMMENTS

Reviewer #1 (Remarks to the Author):

The authors present very interesting and compelling evidence that VCP is the target of SMER28 in activating autophagy-lysosome and UPS pathways for protein degradation. The authors suggest that this compound causes degradation of disease-relevant proteins using HTT and Ataxin 3 repeat-containing cells, and misfolded proteins using puromycin. They show their compound specifically increases the ATPase activity of the D1 ATPase ring of VCP and binds at the N-terminal domain-D1 ATPase domain interface similar to the previously discovered NW1030. This molecule appears to cause VCP to interact more strongly with PI3K complex 1, which stimulates PI(3)P and stimulates autophagy and autophagic flux. Their data suggests that SMER28's effect on the UPS depends on VCP ATPase activity and the presence of cofactors Ufd1 and Npl4. While I find this overall packaging and story very compelling and interesting, I have serious concerns regarding the data as presented. Many of the effect sizes are very small with inappropriate statistical analyses. There also appears to be data repeated across panels without designation, and at least several immunoblot images that have apparent aberrations.

It is interesting that SMER28 binds between the substrate binding domain and the D1 domain. What effect does SMER28 have on gain-of-function VCP mutations associated with multisystem proteinopathy which lie in this approximate region as well? At the very least, it would be helpful to discuss multisystem proteinopathy mutations as these cause VCP to be hyperactive in terms of ATPase activity in vitro and yet cause disease.

Throughout, unpaired t-tests were used for experiments with multiple groups without correction for multiple comparisons. At the very least, an appropriate omnibus ANOVA should be done and reported. This is particularly important given that some of the effect sizes are very small where normalizing to control levels results in reduction in statistical variance

Fig 3i and j show data from what appears to be multiple measurements per experiment (i.e. multiple cells measured for each experiment). An unpaired t-test is not appropriate. In this instance, ANOVA is not appropriate either as measurements are not wholly independent. A mixed effects model or similar should be used. This multiple non-independent measurements issue is also noted for figure 7e, extended data figure 3b, extended data figure 7d where simpler statistical methods are not appropriate.

It is extremely worrisome that data is repeated in different panels. The data points for DMSO+BafA in 3i is the same as the DMSO+BafA in 3j. Similarly the data in 3i for NW1030+BafA is the same as the

NW1030+Baf1 in 3j. While I realize that these are the same experimental conditions, the data is broken into two separate panels with different y-axis and color, presented as if these were different experiments. This is extremely misleading.

This data duplication in figure 3 demonstrates cherry picking in terms of statistical tests. In panel 3i, a t-test is done showing significance with a $p < 0.0005$, but this is not indicated in panel 3j even though the exact same data is displayed.

Another instance of data duplication is figure 3a (SMER28 6h) which is the same as figure 3b (SMER28 alone) where the data from separate panels are identical to each other. These data being presented in different panels with slightly different formatting gives the appearance of an independent replicated experiment. Thus, if this data is duplicated, this practice is very misleading.

Figure 4C has some aberrations which appears that the image was cropped or altered. Similarly, the bottom blots for VCP and VPS34 in figure 4F shows what appears to be some image aberrations where it looks like part of the image was removed and greyed out. An explanation for these image aberrations is required.

Additional comments:

The changes in LC3-II levels in figure 4c (quantified in 4d) are extremely subtle, and if those changes are significant, then it may be that SMER28 decreases LC3-II levels in beclin-KO cells.

I could not find the legends for the extended/supplementary data.

Line 422: Should say "This was not caused..."

Reviewer #2 (Remarks to the Author):

In the current manuscript, Wrobel et. al., broadly explore the strategy of targeting both the ubiquitin-proteasome system and autophagy pathways to clear toxic misfolded/aggregate proteins using a single modulator. Towards this end they use a competitive pull-down approach and found that VCP/p97 is the cellular target for a previously identified small molecule autophagy inducer, SMER28. A previous report by this group found that SMER28 accelerates degradation of harmful neurodegeneration causing protein

species but the target for this small molecule was not known. A limited proteolysis-mass spectrometry approach was used to map the binding site of SMER28 on VCP and identified the cleft formed between N-domain and D1 ATPase domain as a potential binding site. Using biochemical approaches, they show that SMER28 enhances ATPase activity in the D1 but not D2 ATPase domain. They further show that SMER28 activates autophagosome biogenesis by enhancing PI3KC complex assembly and PI(3)P production, in a VCP dependent manner. Intriguingly, SMER28 is not selective for autophagy alone, but can also accelerate proteasomal clearance of toxic protein substrates. Overall, the authors suggest that SMER28 mediated activation of VCP might be an attractive means of treating neurodegenerative diseases.

This has the potential to be an exciting finding, as activators for VCP have been sought after for enabling clearance of aggregates. The authors provide a wealth of well-executed data and it is clear that SMER28 can lead to the clearance of aggregates in cells. However, the evidence provided fails to sufficiently substantiate the conclusions drawn about the effect of SMER28 on VCP-dependent autophagic / proteasomal clearance of misfolded proteins. The major concern is the discrepancy between the very modest increase in VCP D1 ATPase activity in vitro by SMER28 and the significant cellular effects. This leads me to wonder if the mechanism the authors propose is correct. Given the significant number of previous reports that the D2 (and not D1) ATPase activity drives substrate unfolding in VCP, it is difficult to believe that the ~1.2 fold increase in D1 ATPase activity caused by SMER28 is the driver of the significant cellular phenotypes. Given these reservations I do not recommend the manuscript for publication in its current form.

Comments/Concerns:

1. Most of the phenotypes, including, VCP ATPase activity, mutant protein clearance, LC3II conversion were modest but determined to be significant. Unpaired t tests have been used for statistical analyses for the major part of the data analyses even when there are multiple samples to compare. It will be appropriate to use ANOVA with multiple comparisons on the datasets wherever applicable and then determine whether the results are significant.
2. It would be useful to look at the relative contributions of the D1 versus D2 domain using site specific mutants for some of the key cellular assays (PI3P induction, LC3II formation, and PI3KC complex assembly) in presence of SMER28 and/or NW1030. Does SMER28 still augment autophagy and UPS in the cells lacking D1 ATPase activity. This assay was performed in vitro (Fig 2d and e), but given the modest effects, it would be very useful to look at it in the context of cell based assays.
3. In figure 1 they show that treatment with SMER28 reduces polyQ burden by immunoblot. It would be useful to also show fluorescence images and quantify polyQ aggregates +/- SMER28. Is the decrease in signal by immunoblot loss of aggregates or smaller aggregates/ fibrils? Proteostat staining is provided in Fig 1g but it is unclear if there are polyQ aggregates or some other misfolded protein.
4. In supplementary figure 2, the authors develop a series of SMER28 analogs with different functional groups. It is surprising to me that none of these analogs differed from SMER28 (increased or decreased) in terms of activating the D1 ATPase domain of VCP. I wonder again if VCP is in fact the correct target. Can the authors comment?

5. The D1 domain is reported to maintain the hexameric state of VCP. What is the stoichiometry of SMER28 binding to VCP? Since it binds at the cleft between N-domain and the D1 ATPase domain, what is the effect of SMER28 on the hexamer formation? Can the authors rule out that SMER28 does not alter hexamer stability? VCP can exist as tetramers and dodecamer, is SMER28 targeting a higher order VCP assembly?

6. Previous studies have shown that the ATPase rates and unfoldase rates of VCP are not matched especially for disease causing mutations in VCP (Blythe et al Neuron 2019). Can the authors show that the modest increase in D1 ATPase activity is stimulating VCP unfoldase rates to enable clearance? This is a critical experiment in my mind to show that SMER28 mediated activation of VCP enhances substrate unfolding, especially to substantiate the increased degradation of GFP-Ub-G76V by the proteasome in Figure 5.

7. The N-D1 linker of VCP undergoes extensive conformational changes upon adaptor binding, especially with the UFD1-NPL4 heterodimer. Given that the study shows contributions of U-N to SMER28 mediated proteasomal degradation, can the authors study VCP interaction with U-N and other N-D1 adaptors upon SMER28 treatment with purified proteins? Does SMER28 induce an activated conformation of VCP that stimulates adaptor binding? This would suggest an allosteric mechanism akin to NMS873 and not a direct stimulation of D1 ATPase activity.

8. In Figure 3d the SMER28 induced expression of PI3P is blocked by VCP depletion. This is a nice experiment, however, in the supplementary western blot (supp 3a), the depletion of VCP is not complete. Thus, one would expect that SMER28 would still be able to activate residual VCP in these cells and there would be some PI3P production, albeit less than (siCtrl+SMER28)-treated cells. However, in the image and quantification there appears to be no change (compare bars 1, 4,6). This again may suggest that the mechanism the authors propose (increased D1 activation) may not be the only mechanism at play. Can the authors comment?

9. In figure 4f and g, SMER28 is shown to increase assembly of the PI3K complex and DBeQ decreases complex assembly. There is some confusion on how these blots were quantified. There appears to be more ATG14L in the SMER28 treated samples, but the figure legend and methods do not indicate if the increase in ATG14L in the pulldowns was normalized to input. A little clarity on how these fold changes were calculated would be useful. It would also be useful to include a sample where cells are treated concurrently with SMER28 and DBeQ to show that the observed increase in complex assembly is VCP dependent.

10. In Figure 5d, it's not clear why siVCP (in SMER28 treated samples) does not stabilize Ub-G76V more than DMSO alone. This is a well-documented effect of VCP depletion. siVCP alone without SMER28 should also be provided as a control.

11. In Figure 6h and i, they show that the ability of SMER28 to clear puromycin labeled proteins is diminished when UFD1 or NPL4 is depleted. However, the representative blot does not support this claim. Furthermore Ufd1-Npl4 depletion by themselves should increase the puromycin signal. This is not observed in the blot provided and they do not quantify these samples in the graph in i. All lanes should be quantified to see the full extent of differences.

12. In 6l, the representative blot does not appear to show a decrease in puromycin labeled proteins in ATG16L1 KO cells treated with SMER28 (lanes 3 and 4) as the graph in (m) suggests.

13. In Fig 7a, the authors claim that co-localization of VCP with ubiquitin positive inclusions was diminished in NMS and CB treated cells based on line-scans. Manders coefficient is needed here to support this claim. Especially since VCP inhibitors have been shown to stabilize VCP on substrates (Huang, To, et al. MBoC 2018).

Reviewer #3 (Remarks to the Author):

-In this article, the authors identify the target and molecular mechanism of action of SMER28, a positive regulator of autophagy acting via an mTOR-independent mechanism which was shown to prevent the accumulation of amyloid beta-peptide. Using a reverse pull-down competition assay and mass spectrometry, they identified that SEMR28 binds Valosin-containing protein, VCP/p97. They used pharmacology and genetics to further demonstrate that SMER28-mediated induction of autophagy required VCP. They found that SMER also increased PI3P production in a VCP-dependent manner. They identified with great precision the binding site of SMER28 in VCP. They found that SMER28 enhanced both autophagy and proteasome activity. Finally, they showed how SMER induced degradation of both misfolded aggregated and soluble disease-causing proteins such as alpha-synuclein.

-This excellent and original piece of work has major implications for basic knowledge and potential pharmaceutical applications in the field of neurodegenerative diseases.

-The experiments were well carried and controlled, the methodology is sound and the data analysis appeared rigorous, meeting the best standards.

-This reviewer might only suggest being careful in some of the claims for biomedical applications because the extent of the effects, albeit significant, sometimes appeared limited. This reviewer also suggests including representative data from fig S5 in fig 5. In addition, given the role of VCP in ER-phagy, it would be very informative if the authors could test whether or not SMER28 induces ER-phagy.

Reviewer #4 (Remarks to the Author):

SUMMARY

Promoting the elimination of toxic protein species by the proteasome and the autophagy degradation pathway is a prevalent approach to counteract neurodegenerative diseases. Wrobel et al. aim to understand the impact of the small-molecule enhancer SMER28 on VCP, an ATPase which they recently described to stabilize Beclin 1 and in this way induces autophagy (Hill et al. 2021). By introducing a LC3B reporter cell line the authors demonstrate the effect of SMER28 as autophagy inducer and highlight the role of SMER28 in removal of neurotoxic proteins as its treatment decreases levels of mutant Huntingtin and Ataxin-3 but not their wild-type species in fibroblasts. Mass spectrometry data identifies VCP as a target of SMER28, where the binding site for SMER28 is predicted between the substrate binding domain and D1 domain of VCP thereby stimulating VCP D1 but not D2 ATPase activity. Further, SMER28 causes an increased production of PI(3)P which in turn is prevented by inhibition of VCP or VPS34. Different to VCP, SMER28 does not have direct impact on VPS34 kinase activity. By comparing the inducing effects of SMER28 on VCP and PI(3)P production with the previously published VCP activator NW1030, the authors support their mechanism and also provide novel details for NW1030. SMER28- or NW1030-induced autophagy is prevented upon depletion of Beclin 1 and presence of SMER28 results in an enhanced interaction of VCP towards the PI3K complex. Next, the authors show that the enhancing effect of SMER28 on VCP ATPase activity is also linked to protein degradation by the UPS where upstream targeting of substrates but not proteasome activity is affected. This SMER28-induced clearance of misfolded proteins by the UPS remains unaffected when blocking the autophagy degradation pathway and is dependent on the VCP-cofactors UFD1L/NPL4. Finally, the authors reveal that SMER28-mediated removal of soluble neurotoxic proteins is linked to the proteasomal pathway and dependent on VCP's ATPase activity. Taken together, the work of Wrobel et al. uncovers mechanistic details of SMER28 as an enhancer of the VCP D1 ATPase activity and its dual role in promoting autophagy induction and proteasomal degradation simultaneously.

MAJOR COMMENTS

1 In Figure 2a, Log₂ fold change of one replicate is plotted against the second replicate.

This type of graph provides information about the quality control of the experiment but is not representative for statistical evaluation and significance of the data. The authors should show their data in a volcano plot and mention cut-offs of their statistical analysis in the figure legend. As n=2 is a low number for statistical analysis, are these two replicates biological replicates? If not, this experiment should be repeated. In addition, the authors should be more precise with statistical information: a fold change of 2 as it is written in the figure legend (line 1405) equals a log₂ fold change value of 1 on the scale.

2 Did PI4K, IRAK4 or PIK3C show up in the initial MS data set?

3 The authors should include figure legends for all extended data figures.

4 Supplementary tables are missing.

MINOR COMMENTS

5 In lines 235 – 238, the authors already mention that SMER28 stimulates VCP ATPase activity in the D1 domain. However, at this point there is no experimental proof which of the ATPase domains is targeted by SMER28. Please reword this section.

" Compounds activating VCP D1 ATPase enhance both autophagic and proteasomal neurotoxic protein clearance "

We thank Reviewers for their helpful suggestions and comments. We provided new data in Figure 2 f, g; Supplementary Fig. 3g; Supplementary Fig. 5b; Supplementary Fig. 5 j, k.

REVIEWER COMMENTS

Reviewer #1 (Remarks to the Author):

The authors present very interesting and compelling evidence that VCP is the target of SMER28 in activating autophagy-lysosome and UPS pathways for protein degradation. The authors suggest that this compound causes degradation of disease-relevant proteins using HTT and Ataxin 3 repeat-containing cells, and misfolded proteins using puromycin. They show their compound specifically increases the ATPase activity of the D1 ATPase ring of VCP and binds at the N-terminal domain-D1 ATPase domain interface similar to the previously discovered NW1030. This molecule appears to cause VCP to interact more strongly with PI3K complex 1, which stimulates PI(3)P and stimulates autophagy and autophagic flux. Their data suggests that SMER28's effect on the UPS depends on VCP ATPase activity and the presence of cofactors Ufd1 and Npl4.

While I find this overall packaging and story very compelling and interesting, I have serious concerns regarding the data as presented. Many of the effect sizes are very small with inappropriate statistical analyses. There also appears to be data repeated across panels without designation, and at least several immunoblot images that have apparent aberrations.

It is interesting that SMER28 binds between the substrate binding domain and the D1 domain. What effect does SMER28 have on gain-of-function VCP mutations associated with multisystem proteinopathy which lie in this approximate region as well? At the very least, it would be helpful to discuss multisystem proteinopathy mutations as these cause VCP to be hyperactive in terms of ATPase activity in vitro and yet cause disease.

We provide a new experiment in Supplementary Fig. 1i and text in lines 229-232. We purified the multisystem proteinopathy associated mutant VCP-R155H from E.coli and measured its ATPase activity in the presence or absence of SMER28. We also included VCP-WT (wild-type) in this experiment. We observed that VCP-R155H baseline ATPase activity was higher than the WT VCP, in agreement with previously published data. SMER28 treatment significantly increased ATPase activity of the WT VCP (as shown in our previous experiments), but it did not cause significant change in ATPase activity of the disease associated VCP-R155H mutant. These new results confirm that SMER28 is a potent enhancer of VCP ATPase activity.

Throughout, unpaired t-tests were used for experiments with multiple groups without correction for multiple comparisons. At the very least, an appropriate omnibus ANOVA should be done and reported. This is particularly important given that some of the effect sizes are very small where normalizing to control levels results in reduction in statistical variance

In experiments where we compare multiple distinct treatments to a control at the same time/experiment, we have now used an ANOVA or related test. Otherwise, if the perturbations were done at different times, we use t tests and make this clear. We have also added this information to Materials and Methods in Statistics section (line 1154-1162).

In experiments like Fig 2c, where we are testing if inhibitors block the effects of SMER28, we have used t tests for two reasons. First, the experiments all use SMER28, thus samples are not independent. Second, the major part of the experiment is designed to assess if inhibitors block/blunt the increase in ATPase activity caused by SMER28 and therefore we are interested to test whether SMER28 does/does not increase ATPase activity in the presence of diverse inhibitors.

We now provide new statistics analysis:

ANOVA for Fig. 1a, Fig. 2g; Fig. 3 c-e, h, j, m, n; Fig. 4d; Fig. 5d, e; Fig. 6e; Fig. 7f; Supplementary Fig. 1c, g, i; Supplementary Fig. 3c, d, f; Supplementary Fig. 4d; Supplementary Fig. 5m; Supplementary Fig. 6f; Supplementary Fig. 7d, e, h.

Kruskal-Wallis for: Fig 3b; Fig. 7e, h; Supplementary Fig. 4b; Supplementary Fig. 7b.

One sample t test for Fig. 1c-f, h; Fig. 4 g, i; Supplementary Fig. 1d; Supplementary Fig. 3e, h; Supplementary Fig. 4c; Supplementary Fig. 5c, e, f.

Paired two-tailed Student's t test for Fig. 2 c-e; Fig. 3k; Fig. 4a, b, e; Fig. 5a, c, f; Fig. 6 a-c, Fig. 6g-m; Fig. 7b, c, g; Supplementary Fig. 3g; Supplementary Fig. 5h; Supplementary Fig. 6a, b, d, g, i, k; Supplementary Fig. 7g.

Fig 3i and j show data from what appears to be multiple measurements per experiment (i.e. multiple cells measured for each experiment). An unpaired t-test is not appropriate. In this instance, ANOVA is not appropriate either as measurements are not wholly independent. A mixed effects model or similar should be used. This multiple non-independent measurements issue is also noted for figure 7e, Supplementary Fig. 3b, Supplementary Fig. 7d where simpler statistical methods are not appropriate.

We have now moved data from Figure 3i and j into Supplementary Fig. 4b and re-analysed the data using Kruskal-Wallis test with post hoc Dunn's multiple comparison test. We also re-analysed data from Figure 7e and Supplementary Fig. 7d using Kruskal-Wallis test with post hoc Dunn's multiple comparison test. The data in Supplementary Fig. 3b is displayed as Tukey box plot (1.5xIQR), $p > 0.001$ Wilcoxon signed rank test. The changes observed in these experiments are significant.

It is extremely worrisome that data is repeated in different panels. The data points for DMSO+BafA in 3i is the same as the DMSO+BafA in 3j. Similarly the data in 3i for NW1030+BafA is the same as the NW1030+Baf1 in 3j. While I realize that these are the same experimental conditions, the data is broken into two separate panels with different y-axis and color, presented as if these were different experiments. This is extremely misleading. This data duplication in figure 3 demonstrates cherry picking in terms of statistical tests. In panel 3i, a t-test is done showing significance with a $p < 0.0005$, but this is not indicated in panel 3j even though the exact same data is displayed.

Thank you for pointing this out. We split the data in order to make the points more clear and to show a logical order in the text. We have now included only the graph showing no additive affect for SMER28 and NW1030 for ability to induce autophagy (measured by LC3 puncta number). Data in Supplementary Fig. 4b. To analyse this data we used Kruskal-Wallis test with post hoc Dunn's multiple comparison tests showing significant difference to DMSO treated samples.

Another instance of data duplication is figure 3a (SMER28 6h) which is the same as figure 3b (SMER28 alone) where the data from separate panels are identical to each other. These data being presented in different panels with slightly different formatting gives the appearance of an independent replicated experiment. Thus, if this data is duplicated, this practice is very misleading.

Thank you for this point and we apologise for the mistake. The data showing the increase in PI(3)P upon SMER28 treatment which belongs to the experiment in Figure 3b was erroneously put on the graph in Figure 3j. This has been now corrected and it does not change the final conclusions from this experiment. To clarify presentation and since the SMER28 and EBSS were done at different times the data are now separated in Figs 3b and 3c.

Figure 4C has some aberrations which appears that the image was cropped or altered. Similarly, the bottom blots for VCP and VPS34 in figure 4F shows what appears to be some image aberrations where it looks like part of the image was removed and greyed out. An explanation for these image aberrations is required.

In order to analyse the levels of multiple proteins for each sample condition, after proteins are resolved on SDS-PAGE gels and transferred to PVDF membrane, the membrane is cut and incubated with different antibodies. We provided better blot in Figure 4c. In Figure 4F the cuts were made close to the bands which likely causes these aberrations – but this was done so we could analyse all components of PI3K complex on one gel. We provide an explanation in the figure legend for Figure 4 and in Materials and Methods in Western blot section (line 748-750). Additionally please see below the corresponding whole cropped membrane.

Additional comments:

The changes in LC3-II levels in figure 4c (quantified in 4d) are extremely subtle, and if those changes are significant, then it may be that SMER28 decreases LC3-II levels in beclin-KO cells.

We have now put a more representative LC3 and corresponding Actin blot in Figure 4c and re-analysed the data from Figure 4c (quantified in 4d) using ANOVA with post hoc Tukey test and SMER28 significantly increases LC3-II upon SMER28. However, it seems to have opposite effect in Beclin KO cells, showing a very small but significant decrease upon SMER28 treatment. This Beclin1-dependence on the autophagy-inducing effects of SMER28 is reproducible and is consistent with our model. Also please note that we show autophagy flux is increased with SMER28 using multiple assays, including Figure 1a.

I could not find the legends for the Supplementary/Supplementary. Apologies if these were not obvious on the journal webpage.

Supplementary Figure legends are included.

Line 422: Should say "This was not caused..."

Apologies, we have not corrected this as we could not identify the relevant text.

Reviewer #2 (Remarks to the Author):

In the current manuscript, Wrobel et. al., broadly explore the strategy of targeting both the ubiquitin-proteasome system and autophagy pathways to clear toxic misfolded/aggregate proteins using a single modulator. Towards this end they use a competitive pull-down approach and found that VCP/p97 is the cellular target for a previously identified small molecule autophagy inducer, SMER28. A previous report by this group found that SMER28 accelerates degradation of harmful neurodegeneration causing protein species but the target for this small molecule was not known. A limited proteolysis-mass spectrometry approach was used to map the binding site of SMER28 on VCP and identified the cleft formed between N-domain and D1 ATPase domain as a potential binding site. Using biochemical approaches, they show that SMER28 enhances ATPase activity in the D1 but not D2 ATPase domain. They further show that SMER28 activates autophagosome biogenesis by enhancing PI3KC complex assembly and PI(3)P production, in a VCP dependent manner. Intriguingly, SMER28 is not selective for autophagy alone, but can also accelerate proteasomal clearance of toxic protein substrates. Overall, the authors suggest that SMER28 mediated activation of VCP might be an attractive means of treating neurodegenerative diseases.

This has the potential to be an exciting finding, as activators for VCP have been sought after for enabling clearance of aggregates. The authors provide a wealth of well-executed data and it is clear that SMER28 can lead to the clearance of aggregates in cells. However, the evidence provided fails to sufficiently substantiate the conclusions drawn about the effect of SMER28 on VCP-dependent autophagic / proteasomal clearance of misfolded proteins. The major concern is the discrepancy between the very modest increase in VCP D1 ATPase activity in vitro by SMER28 and the significant cellular effects. This leads me to wonder if the mechanism the authors propose is correct. Given the significant number of previous reports that the D2 (and not D1) ATPase activity drives substrate unfolding in VCP, it is difficult to believe that the ~1.2 fold increase in D1 ATPase activity caused by SMER28 is the driver of the significant cellular phenotypes.

Our data suggest that modest increases in VCP D1 ATPase activity correlate with enhanced autophagosome formation and non-autophagic proteasome-dependent clearance. We show data with two compounds that bind VCP and stimulate VCP D1 activity, and show the effects are VCP dependent using both chemical and genetic approaches. (Fig. 3c-e, h, j, k, m, n; Fig. 4c-e; Fig. 5c-e; Fig. 6f-l; Fig. 7 a-c, h; Supplementary Fig.2; Supplementary Fig. 3 b, e, f; Supplementary Fig. 4 d-g; Supplementary Fig. 7c). However, we agree that we cannot easily address whether the protein clearance effects of SMER28 are specifically or exclusively due to enhanced D1 ATPase activity. For example, SMER28 may have allosteric effects on VCP which result in the protein clearance enhancement and the modest increase in D1 ATPase activity may be an epiphenomenon. We have now added text to the discussion (line 559-563) and modified the abstract (line 43-45) to ensure that this possibility is explicit. We have changed the title accordingly too (Compounds activating VCP D1 ATPase enhance both autophagic and proteasomal neurotoxic protein clearance).

Comments/Concerns:

1. Most of the phenotypes, including, VCP ATPase activity, mutant protein clearance, LC3II conversion were modest but determined to be significant. Unpaired t tests have been used for statistical analyses for the major part of the data analyses even when there are multiple samples to compare. It will be appropriate to use ANOVA with multiple comparisons on the datasets wherever applicable and then determine whether the results are significant.

In experiments where we compare multiple distinct treatments to a control at the same time/experiment, we have now used an ANOVA or related test. Otherwise, if the perturbations were done at different times, we use t tests and make this clear. We have also added this information to Materials and Methods in Statistics section (line 1154-1162).

In experiments like Fig 2c, where we are testing if inhibitors block the effects of SMER28, we have used t tests for two reasons. First, the experiments all use SMER28, thus lanes are not independent. Second, the major part of the experiment is designed to assess if inhibitors block/blunt the increase in ATPase activity caused by SMER28 so we are looking to test if the SMER does/does not increase ATPase activity in the presence of diverse inhibitors.

We now provide new statistics analysis:

ANOVA for Fig. 1a, Fig. 2g; Fig. 3 c-e, h, j, m, n; Fig. 4d; Fig. 5d, e; Fig. 6e; Fig. 7f; Supplementary Fig. 1c, g, i; Supplementary Fig. 3c, d, f; Supplementary Fig. 4d; Supplementary Fig. 5m; Supplementary Fig. 6f; Supplementary Fig. 7d, e, h.

Kruskal-Wallis for: Fig 3b; Fig. 7e, h; Supplementary Fig. 4b; Supplementary Fig. 7b.

One sample t test for Fig. 1c-f, h; Fig. 4 g, i; Supplementary Fig. 1d; Supplementary Fig. 3e, h; Supplementary Fig. 4c; Supplementary Fig. 5c, e, f.

Paired two-tailed Student's t test for Fig. 2 c-e; Fig. 3k; Fig. 4a, b, e; Fig. 5a, c, f; Fig. 6 a-c, Fig. 6g-m; Fig. 7b, c, g; Supplementary Fig. 3g; Supplementary Fig. 5h; Supplementary Fig. 6a, b, d, g, i, k; Supplementary Fig. 7g.

2. It would be useful to look at the relative contributions of the D1 versus D2 domain using site specific mutants for some of the key cellular assays (PI3P induction, LC3II formation, and PI3KC complex assembly) in presence of SMER28 and/or NW1030. Does SMER28 still augment autophagy and UPS in the cells lacking D1 ATPase activity. This assay was performed in vitro (Fig 2d and e), but given the modest effects, it would be very useful to look at it in the context of cell based assays.

The proposed experiment are likely almost impossible to pursue. VCP knockouts are lethal and even transient knockdowns are toxic for cells. There are no available VCP inhibitors which target only the D1 domain. We found that overexpression of VCP with D1 or D2 loss-of-function mutations, even in cells which still have endogenous wild-type VCP, is toxic for cells (unpublished observations from our lab) suggesting that attempt to create cell line with endogenous point mutation in D1 or D2 domain of VCP which will abolish ATPase activity will fail as the cells will be not viable. Thus, in our opinion the only way to test D1 versus D2 dependence of SMER28 activation is using in vitro assays, which we have done (Fig. 2c-e, Supplementary Fig. 1g, h, i; Supplementary Fig. 2e).

3. In figure 1 they show that treatment with SMER28 reduces polyQ burden by immunoblot. It would be useful to also show fluorescence images and quantify polyQ aggregates +/- SMER28. Is the decrease in signal by immunoblot loss of aggregates or smaller aggregates/ fibrils? Proteostat staining is provided in Fig 1g but it is unclear if there are polyQ aggregates or some other misfolded protein.

The levels of mutant mHTT aggregates upon SMER28 treatment were extensively analysed in our previous publication (Sarkar, 2007; PMID: 17486044) in COS-7 and MEF cells and they were decreased upon SMER28 in autophagy-dependent manner (ATG5 null cells used). It is possible that the immunoblot represents fibrils/smaller aggregates which were soluble in Laemmli buffer and could enter the gel. These species could be targeted by autophagy and/or ubiquitin proteasome system.

4. In Supplementary Fig. 2, the authors develop a series of SMER28 analogs with different functional groups. It is surprising to me that none of these analogs differed from SMER28 (increased or decreased) in terms of activating the D1 ATPase domain of VCP. I wonder again if VCP is in fact the correct target. Can the authors comment?

Sorry this is not correct - unlike SMER28 and some other analogs, analogs B and C do not change VCP ATPase activity (Supplementary Fig.2 e) and this is reflected in their lack of ability to induce autophagy (measured by LC3 puncta in Supplementary Fig.2 f, g) or increase degradation of mutant alpha-synuclein (Supplementary Fig. 2 h). Moreover, analogs B and C are not able to enhance degradation of proteasome reporter substrate Ub-G76V-GFP (Fig. 5 b). In our opinion, experiments using SMER28 analogs are key to allow us to correlate SMER28 effects on VCP ATPase activity with its effects on autophagy and UPS.

5. The D1 domain is reported to maintain the hexameric state of VCP. What is the stoichiometry of SMER28 binding to VCP? Since it binds at the cleft between N-domain and the D1 ATPase domain, what is the effect of SMER28 on the hexamer formation? Can the authors rule out that SMER28 does not alter hexamer stability? VCP can exist as tetramers and dodecamer, is SMER28 targeting a higher order VCP assembly?

We provide a new experiment in Fig. 2f, quantification in Fig. 2g and text in lines 232-237. We treated HeLa cells with SMER28 or VCP inhibitor for 6 h, prior to sample analysis using native gel conditions. As expected, VCP ATPase activity inhibitor (CB-5083) significantly decreased the levels of the VCP hexamer. However, SMER28 did not cause any significant change in the levels of VCP hexamer. Moreover, it does not seem to alter the levels of the higher order VCP assemblies, although the signal detected on the blot is very weak.

Regarding the stoichiometry of SMER28 binding to VCP, the LiP assay is not designed to assess the stoichiometry of binding (unless one considers the computed IC50 a proxy for it). If SMER28 were to disrupt the hexamers or bind preferentially to other oligomeric complexes (tetramer or dodecamer) we would expect to detect high correlation LiP peptides from other regions of the protein but we do not observe this. Although this is not concrete proof that these other binding events do not occur (we cannot rule them out completely) it is strong evidence that if they do occur, they are either the minority of binding events, or SMER28 binds in the same location in those other formations.

6. Previous studies have shown that the ATPase rates and unfoldase rates of VCP are not matched especially for disease causing mutations in VCP (Blythe et al Neuron 2019). Can the authors show that the modest increase in D1 ATPase activity is stimulating VCP unfoldase rates to enable clearance? This is a critical experiment in my mind to show that SMER28 mediated activation of VCP enhances substrate unfolding, especially to substantiate the increased degradation of GFP-Ub-G76V by the proteasome in Figure 5.

The ATPase activity of D2 domain was shown to be a main pulling force for incoming substrates (PMID: 28475898) and therefore it is likely that the change in D1 ATPase activity (but not in D2) caused by SMER28 binding will not influence the unfoldase rate. There are few possibilities which could be tested in following studies. First, it is possible that the increase in D1 ATPase activity could modulate substrate recruitment, as D1 ATPase hydrolysis seems to change the confirmation of N-domain which changes the binding surface on VCP (PMID: 31249135; PMID: 31249134). Second, it is also possible the changes in the D1 ATPase activity could influence recruitment of cofactors, other than NPL4/UFD1L, as we showed their binding to VCP is not significantly affected upon SMER28 treatment (please see new Supplementary Fig. 5j-k). However, such studies are clearly beyond the scope of this paper.

7. The N-D1 linker of VCP undergoes extensive conformational changes upon adaptor binding, especially with the UFD1-NPL4 heterodimer. Given that the study shows contributions of U-N to SMER28 mediated proteasomal degradation, can the authors study VCP interaction with U-N and other N-D1 adaptors upon SMER28 treatment with purified proteins? Does SMER28 induce an activated conformation of VCP that stimulates adaptor binding? This would suggest an allosteric mechanism akin to NMS873 and not a direct stimulation of D1 ATPase activity.

We provide new experiments in Supplementary Fig. 5j, k and text in lines 421-426. We tested UFD1-NPL4 binding with VCP in the presence of SMER28 using an *in vitro* and in cells approach. We observed that SMER28 does not cause a significant change in UFD1-NPL4 binding to VCP, even when we increased SMER28 concentration in vitro up to 50 μ M. Thus, the observed increased rate of UPS substrate degradation upon SMER28 (for example with the Ub-G76V-GFP reporter) cannot be explained by increased U/N adaptors binding. However, we still cannot exclude that SMER28 induces some allosteric changes to VCP, in addition to increasing its ATPase activity, which contribute to observed phenotypes. This has been added in Discussion, please see text in lines 562-566.

8. In Figure 3d the SMER28 induced expression of PI3P is blocked by VCP depletion. This is a nice experiment, however, in the Supplementary western blot (supp 3a), the depletion of VCP is not complete. Thus, one would expect that SMER28 would still be able to activate residual VCP in these cells and there would be some PI3P production, albeit less than (siCtrl+SMER28)-treated cells. However, in the image and quantification there appears to be no change (compare bars 1, 4,6). This again may suggest that the mechanism the authors propose (increased D1 activation) may not be the only mechanism at play. Can the authors comment?

In this experiment, VCP is not completely depleted although the knockdown is pretty efficient – please note that complete lack of VCP is lethal. However, the decrease is substantial enough to see that the SMER28 could not increase PI3P in these cells to the levels observed in the wild-type cells. It is possible that any residual VCP by SMER28 is not sufficient to cause a detectable increase in PI(3)P signal, as the levels of the lipid need to be sufficient to allow a clear signal from the anti-PI3P antibody.

9. In figure 4f and g, SMER28 is shown to increase assembly of the PI3K complex and DBeQ decreases complex assembly. There is some confusion on how these blots were quantified. There appears to be more ATG14L in the SMER28 treated samples, but the figure legend and methods do not indicate if the increase in ATG14L in the pulldowns was normalized to input. A little clarity on how these fold changes were calculated would be useful. It would also be useful to include a sample where cells are treated concurrently with SMER28 and DBeQ to show that the observed increase in complex assembly is VCP dependent.

The data analysis for IP experiment was done by normalising immunoprecipitated protein to the bait (in this case ATG14L) in the IP fraction. The data are presented as normalised to DMSO levels. We provide this information in the figure legends.

We have now provided a new experiment in Supplementary Fig. 5b and text in lines 384-386. In this experiment, we confirm that increase in the PI3K complex assembly upon SMER28 treatment is VCP-dependent as it was abolished by the VCP inhibitor DBeQ.

10. In Figure 5d, it's not clear why siVCP (in SMER28 treated samples) does not stabilize Ub-G76V more than DMSO alone. This is a well-documented effect of VCP depletion. siVCP alone without SMER28 should also be provided as a control.

Apologies about the confusion. In Fig. 5d, we have normalised the control data in the absence/presence of VCP knockdown to 1, so that we can compare the magnitude of the effects of SMER28 on the proteasome substrate. The effect of the VCP knockdown alone is shown in Supplementary Fig. 5h and is now explicitly cross-referenced in the figure legend of Fig. 5d (lines 1532-1536).

11. In Figure 6h and i, they show that the ability of SMER28 to clear puromycin labeled proteins is diminished when UFD1 or NPL4 is depleted. However, the representative blot does not support this claim. Furthermore Ufd1-Npl4 depletion by themselves should increase the puromycin signal. This is not observed in the blot provided and they do not quantify these samples in the graph in i. All lanes should be quantified to see the full extent of differences.

We provide new data quantification for the levels of puromycin-labelled proteins in Figure 6 h, i. We observe a decrease in the level of puromycin-labelled proteins upon knockdown of NPL4 or UFD1L, which is possibly caused by the overall decrease in the protein synthesis rate in these cells (experimental observations in our lab, data not shown). However, when measuring the levels of puromycin-labelled proteins in DMSO- and SMER28-treated conditions in wild-type and knockdown NPL4 or UFD1L cells, we confirm that NPL4 and UFD1L are necessary for SMER28-mediated degradation of puromycin-labelled proteins.

12. In 6l, the representative blot does not appear to show a decrease in puromycin labeled proteins in ATG16L1 KO cells treated with SMER28 (lanes 3 and 4) as the graph in (m) suggests.

We have now provided a more representative blot for this experiment in Fig. 6l.

13. In Fig 7a, the authors claim that co-localization of VCP with ubiquitin positive inclusions was diminished in NMS and CB treated cells based on line-scans. Manders coefficient is needed here to support this claim. Especially since VCP inhibitors have been shown to stabilize VCP on substrates (Huang, To, et al. MBoC 2018).

We provided a new analysis of data in Supplementary Fig. 7b. We used Manders' coefficient analysis which fully supported our initial observation that in cells treated with NMS873 or CB-5083 we could detect decreased localisation of VCP with UB-positive inclusions.

Reviewer #3 (Remarks to the Author):

-In this article, the authors identify the target and molecular mechanism of action of SMER28, a positive regulator of autophagy acting via an mTOR-independent mechanism which was shown to prevent the accumulation of amyloid beta-peptide. Using a reverse pull-down competition assay and mass spectrometry, they identified that SEMR28 binds Valosin-containing protein, VCP/p97. They used pharmacology and genetics to further demonstrate that SMER28-mediated induction of autophagy required VCP. They found that SMER also increased PI3P production in a VCP-dependent manner. They identified with great precision the binding site of SMER28 in VCP. They found that SMER28 enhanced both autophagy and proteasome activity. Finally, they showed how SMER induced degradation of both misfolded aggregated and soluble disease-causing proteins such as alpha-synuclein.

-This excellent and original piece of work has major implications for basic knowledge and potential pharmaceutical applications in the field of neurodegenerative diseases.

-The experiments were well carried and controlled, the methodology is sound and the data analysis appeared rigorous, meeting the best standards.

-This reviewer might only suggest being careful in some of the claims for biomedical applications because the extent of the effects, albeit significant, sometimes appeared limited. This reviewer also suggests including representative data from fig S5 in fig 5. In addition, given the role of VCP in ER-phagy, it would be very informative if the authors could test whether or not SMER28 induces ER-phagy.

We provided a new experiment in Supplementary Fig. 3g and text in lines 306-312. To address this question, we used a well-established marker of the ER-phagy – the ER receptor TEX264 (PMID: 31006538, PMID: 31006537). Treatment with SMER28 induced clearance of the general autophagy receptor p62, but it did not change the levels of ER-phagy specific receptor TEX264 at the 9 h of treatment. At the same time starvation induced the clearance of both receptors as expected (Supplementary Fig. 3g). As the EBSS treatment induces autophagy much faster, we cannot exclude that an effect on ER-phagy may be observed at much later time points than what we used in the assays in this paper.

Reviewer #4 (Remarks to the Author):

SUMMARY

Promoting the elimination of toxic protein species by the proteasome and the autophagy degradation pathway is a prevalent approach to counteract neurodegenerative diseases. Wrobel et al. aim to understand the impact of the small-molecule enhancer SMER28 on VCP, an ATPase which they recently described to stabilize Beclin 1 and in this way induces autophagy (Hill et al. 2021). By introducing a LC3B reporter cell line the authors demonstrate the effect of SMER28 as autophagy inducer and highlight the role of SMER28 in removal of neurotoxic proteins as its treatment decreases levels of mutant Huntingtin and Ataxin-3 but not their wild-type species in fibroblasts. Mass spectrometry data identifies VCP as a target of SMER28, where the binding site for SMER28 is predicted between the substrate binding domain and D1 domain of VCP thereby stimulating VCP D1 but not D2 ATPase activity. Further, SMER28 causes an increased production of PI(3)P which in turn is prevented by inhibition of VCP or VPS34. Different to VCP, SMER28 does not have direct impact on VPS34 kinase activity. By comparing the inducing effects of SMER28 on VCP and PI(3)P production with the previously published VCP activator NW1030, the authors support their mechanism and also provide novel details for NW1030. SMER28- or NW1030-induced autophagy is prevented upon depletion of Beclin 1 and presence of SMER28 results in an enhanced interaction of VCP towards the PI3K complex. Next, the authors show that the enhancing effect of SMER28 on VCP ATPase activity is also linked to protein degradation by the UPS where upstream targeting of substrates but not proteasome activity is affected. This SMER28-induced clearance of misfolded proteins by the UPS remains unaffected when blocking the autophagy degradation pathway and is dependent on the VCP-cofactors UFD1L/NPL4. Finally, the authors reveal that SMER28-mediated removal of soluble neurotoxic proteins is linked to the proteasomal pathway and dependent on VCP's ATPase activity. Taken together, the work of Wrobel et al. uncovers mechanistic details of SMER28 as an enhancer of the VCP D1 ATPase activity and its dual role in promoting autophagy induction and proteasomal degradation simultaneously.

MAJOR COMMENTS

1 In Figure 2a, Log₂ fold change of one replicate is plotted against the second replicate. This type of graph provides information about the quality control of the experiment but is not representative for statistical evaluation and significance of the data. The authors should show their data in a volcano plot and mention cut-offs of their statistical analysis in the figure legend. As n=2 is a low number for statistical analysis, are these two replicates biological replicates? If not, this experiment should be repeated. In addition, the authors should be more precise with statistical information: a fold change of 2 as it is written in the figure legend (line 1405) equals a log₂ fold change value of 1 on the scale.

The graph represents two biological replicates, which in our opinion were enough to get initial confidence in the proteins identified. The Mass Spec analysis of proteins which bound to the SMER28-linked beads was our initial experiment which allow us to identify the strongest SMER28 binding protein – VCP. We further validated our data with multiple *in vitro* and in cells assays which fully support our initial identification that the main target of SMER28 is VCP. We changed the text in figure legends as suggested (lines 1436-1438).

2 Did PI4K, IRAK4 or PIK3C show up in the initial MS data set?

PI4K and IRAK4 were not detected in the initial MS data. PIK3C was detected to bind SMER28, however with very low affinity. For more information please see Supplementary Table 1.

3 The authors should include figure legends for all Supplementary figures.

We provided figure legends to Supplementary. Apologies if these were not obvious on the journal webpage.

4 Supplementary tables are missing.

We provided 3 Supplementary tables. Apologies if these were not obvious on the journal webpage.

MINOR COMMENTS

5 In lines 235 – 238, the authors already mention that SMER28 stimulates VCP ATPase activity in the D1 domain. However, at this point there is no experimental proof which of the ATPase domains is targeted by SMER28. Please reword this section.

We changed the order in the text to avoid misleading readers (lines 219-238).

REVIEWER COMMENTS

Reviewer #1 (Remarks to the Author):

The authors have been very receptive to several aspects of the prior critique. In particular, addressing the effects of SMER28 on mutant VCP, and exploring its effects on cofactor binding add important details to this manuscript. While I find the results highly interesting, I still have concerns over the statistical analysis of the data which I try to explain with some detail here. My emphasis on the statistical methods is to hopefully ascertain the significance of the relatively small effect sizes with more rigor. As such, several points raised previously were not fully addressed.

As mentioned previously, normalizing to control levels results in a reduction in statistical variance. There are even instances where different groups are used for normalization such as figure 4d where the DMSO group is used for control cells but the SMER28 group is used for BECLIN1 KO cells. This lack of homoscedasticity violates the basic assumptions that underlie t-test and ANOVA.

There are also still instances of data where an ANOVA is not used at all in lieu of multiple t-tests of selected pairs. The authors argue that these instances of using multiple t-tests represented different experiments done at different times. If so, the data should not be presented together as one graph to emphasize that these were independent experiments. However, the authors also describe their rationale for using t-tests highlighting figure 2c – but looking at this data, the entire dataset is normalized to the DMSO control so it is presented as a single experiment done together.

In terms of t-tests (ignoring the problem of unequal variance described above), there is an instance of using a one-tailed t-tests do not seem appropriate. In other instances, a one-sample t-test is used while other datasets are analyzed with paired t-tests. The rationale for these different choices is not clear.

The prior critique raised the issue of multiple non-independent measurements being analyzed by ANOVA. In this instance, changing to a Kruskal-Wallis test is not appropriate in lieu of a model that takes into account multiple repeated measures.

Overall, the statistical analysis is variable and often violates basic statistical assumptions. Analyzing the data in a statistically rigorous manner would greatly add to this study.

Reviewer #2 (Remarks to the Author):

In the revised version of the manuscript, Wrobel et. al., have addressed most of this reviewer's comments with supporting arguments. This has improved the manuscript and the data support the conclusions.

However, the suggested experiment in the previous review (comment#2) regarding assessing the activity of SMER28 in cells expressing the D1 mutant should be attempted. VCP knockdown and rescue with wildtype p97, D1, and D2 ATPase activity deficient point mutants have been performed by many groups in various publications in HeLa cells suggesting that the problems with cell toxicity can be overcome. Moreover the authors themselves have knocked down VCP using siRNA in HeLa cells to quantify PI3P levels without apparent cellular toxicity (Figure 3E).

Reviewer #3 (Remarks to the Author):

The authors have satisfactorily answered the reviewers' comments.

Reviewer #4 (Remarks to the Author):

The authors adequately addressed all my critical points. Therefore, I recommend to accept this manuscript for publication.

Reviewer #1 (Remarks to the Author):

The authors have been very receptive to several aspects of the prior critique. In particular, addressing the effects of SMER28 on mutant VCP, and exploring its effects on cofactor binding add important details to this manuscript. While I find the results highly interesting, I still have concerns over the statistical analysis of the data which I try to explain with some detail here. My emphasis on the statistical methods is to hopefully ascertain the significance of the relatively small effect sizes with more rigor. As such, several points raised previously were not fully addressed.

As mentioned previously, normalizing to control levels results in a reduction in statistical variance. There are even instances where different groups are used for normalization such as figure 4d where the DMSO group is used for control cells but the SMER28 group is used for BECLIN1 KO cells. This lack of homoscedasticity violates the basic assumptions that underlie t-test and ANOVA.

We previously consulted with a professional statistician about the problem that cell biologists have to confront when analysing experiments performed on different days. For example, with western blots of such independent experiments, the value of the band of interest and the loading control each vary according to loading and gels exposure and other factors like antibody concentrations. (Indeed, even the same sample loaded on different gels will give different raw values). Our advisor (David Clayton) suggested that we could normalise control values to 1 and then perform one-sample t tests (where these involved one perturbation).

The approach we have used is considered in detail in <https://www.ncbi.nlm.nih.gov/pmc/articles/PMC3903630/>. This article nicely explains why normalisation is required. Note that we have used a LICOR machine to quantify which is more linear than ECL blotting. The normalisation approach we have used if anything is more conservative, as it reduces false positives and increases false negatives. Also note that an analogous approach is required for immunofluorescence of cells and related analyses.

We are sorry if the reviewer misinterpreted Fig 4d - in this experiment, LC3-II levels were normalized to DMSO control in BECLIN1 control cells. Please see raw data for this experiment in Source Data excel file.

There are also still instances of data where an ANOVA is not used at all in lieu of multiple t-tests of selected pairs. The authors argue that these instances of using multiple t-tests represented different experiments done at different times. If so, the data should not be presented together as one graph to emphasize that these were independent experiments. However, the authors also describe their rationale for using t-tests highlighting figure 2c – but looking at this data, the entire dataset is normalized to the DMSO control so it is presented as a single experiment done together.

As the experiments were indeed performed at different times, we have now presented the VCP ATPase activity upon SMER28 or with VCP inhibitors treatment as separate graphs in Figure 2. We have analysed the data from SMER28-treated sample with one sample t test and conditions in which SMER28 was combined with VCP inhibitors with ANOVA with post hoc Tukey test.

In terms of t-tests (ignoring the problem of unequal variance described above), there is an instance of using a one-tailed t-tests do not seem appropriate. In other instances, a one-sample t-test is used while other datasets are analyzed with paired t-tests. The rationale for these different choices is not clear.

The only time we use a one-tailed t test is in Fig 3L for analysing the change in the number of WIPI2 and ATG16 puncta upon autophagy inducing conditions (EBSS). The reason we do this is that it is well established that WIPI2 and ATG16 puncta should increase upon starvation (EBSS conditions), so in our opinion one-tail t-test is appropriate to analyse the data comparing control and EBSS conditions, since the direction of change is established.

Following Reviewer's suggestion, we have now changed statistical analysis methods in the following panels:

- In Figure 3g - changed to paired two-tailed Student's t-test.
- In Figure 4e - changed for one sample t test.
- In Figure 6 g, k, m - changed for one sample t test.
- In Figure 7c - changed to one sample t test.

The prior critique raised the issue of multiple non-independent measurements being analyzed by ANOVA. In this instance, changing to a Kruskal-Wallis test is not appropriate in lieu of a model that takes into account multiple repeated measures.

We have now removed the representative graph from Figure 7e and left the statistical analysis of three independent biological replicates (now Figure 7e). In this experiment we have normalized the SMER28-treated conditions to DMSO for WT and ATG16 KO cells separately as this allows us to compare the SMER28/DMSO ratios for WT and ATG16 KO cell. Note that the question being asked is whether the extent to which ubiquitinated inclusions are lowered in wild-type cells is more than in the autophagy null-cells (comparison of 2nd and 4th bars).

We have also removed a representative distribution graph from Supplementary Figure 7b and replaced it with statistical analysis of three independent biological replicates. This does not change our final conclusions from this experiment.

Overall, the statistical analysis is variable and often violates basic statistical assumptions. Analyzing the data in a statistically rigorous manner would greatly add to this study.

Please see comments above.

Reviewer #2 (Remarks to the Author):

In the revised version of the manuscript, Wrobel et. al., have addressed most of this reviewer's comments with supporting arguments. This has improved the manuscript and the data support the conclusions.

However, the suggested experiment in the previous review (comment#2) regarding assessing the activity of SMER28 in cells expressing the D1 mutant should be attempted. VCP knockdown and rescue with wildtype p97, D1, and D2 ATPase activity deficient point mutants have been performed by many groups in various publications in HeLa cells suggesting that the problems with cell toxicity can be overcome. Moreover the authors themselves have knocked down VCP using siRNA in HeLa cells to quantify PI3P levels without apparent cellular toxicity (Figure 3E).

The reviewer says that experiment with expression of D1 and D2-VCP mutants were performed in the past by various groups. However, many of these studies observed extensive cellular toxicity caused by expression of VCP D1 or especially D2 mutants.

"It is important to note that high expression levels of some p97/VCP mutants (particularly those in the D2 domain) were not well tolerated for these assays, exhibiting lethality in some transfected cells." "Even though p97/VCPE305Q expression impaired ERAD to an extent comparable to that obtained with other Walker mutations, cells appeared to tolerate higher expression levels of this mutant (Figure S5), in agreement with results observed previously" (PMID: 16713576).

"expression of p97(E578Q) causes ubiquitinated proteins to accumulate on ER membranes and slows degradation of the ERAD substrate cystic-fibrosis transmembrane-conductance regulator. In addition, expression of p97(E578Q) eventually causes the ER to swell." (PMID: 14617820)

We have also measured a cellular toxicity upon overexpression of VCP wild-type (WT), D1 or D2 mutants using LDH assay (see graph below) and clearly observed that presence of D1 or D2 mutant in cells caused an extensive cellular toxicity and death. This does not allow us to perform meaningful autophagy experiments (since caspase activity inhibits Beclin 1-dependent autophagy (PMID: 19713971). Thus, the experiment that this reviewer has requested is not feasible. However, note that we have provided in vitro support for the relevance of the D1 ATPase activity (Fig 2).

REVIEWERS' COMMENTS

Reviewer #1 (Remarks to the Author):

I appreciate the authors' clarification that in many instances a one-sample t-test is used which was not evident to me previously. I also appreciate having the source data document and more explicit figure legends so that it is transparent what test is done in each case.

A few minor mistakes are found where the source data document and figure legends for figure 2E, sup fig 6g, 6i and 6k and sup fig 3h indicates a one sample t-test was used although it looks like a paired t-tests were done. These errors can be corrected with very minor edits.

There are a few instances (fig 5b, sup 2c,d,e,g,h, sup 3g) where unpaired two-sample t-tests are still used. Also, there is no equivalent of a "one sample ANOVA" to deal with the lack of variance in the control group and so the use of ANOVA throughout is still not ideal. I also disagree that the instance of using a one-tailed t-test is appropriate (fig 3l) but I realize that this is a matter of opinion. That being said, I also appreciate that statistics should not be the only basis for evaluating data. Therefore, given that the numerical values are all available in the source data document, the overall analysis is satisfactory. I appreciate the authors' willingness to adjust some of their analyses.

Reviewer #2 (Remarks to the Author):

The authors have satisfactorily addressed my concerns.

Nature Communications manuscript NCOMMS-21-40321B

Compounds activating VCP D1 ATPase enhance both autophagic and proteasomal neurotoxic protein clearance

Reviewer #1 (Remarks to the Author):

I appreciate the authors' clarification that in many instances a one-sample t-test is used which was not evident to me previously. I also appreciate having the source data document and more explicit figure legends so that it is transparent what test is done in each case.

A few minor mistakes are found where the source data document and figure legends for figure 2E, sup fig 6g, 6i and 6k and sup fig 3h indicates a one sample t-test was used although it looks like a paired t-tests were done. These errors can be corrected with very minor edits.

Thank you for your comments. We have now corrected the mistake in Source Data file.

There are a few instances (fig 5b, sup 2c,d,e,g,h, sup 3g) where unpaired two-sample t-tests are still used. Also, there is no equivalent of a "one sample ANOVA" to deal with the lack of variance in the control group and so the use of ANOVA throughout is still not ideal. I also disagree that the instance of using a one-tailed t-test is appropriate (fig 3l) but I realize that this is a matter of opinion. That being said, I also appreciate that statistics should not be the only basis for evaluating data.

Therefore, given that the numerical values are all available in the source data document, the overall analysis is satisfactory. I appreciate the authors' willingness to adjust some of their analyses.

Thanks for being flexible on this statistical issue.